# PARP14 is regulated by the PARP9/DTX3L complex and promotes interferon γ-induced ADP-ribosylation

Victoria Chaves Ribeiro [ID][1,2], Lilian Cristina Russo [ID][1,2] & Nícolas Carlos Hoch [ID][1✉]

## Abstract

**Protein ADP-ribosylation plays important but ill-defined roles in antiviral signalling cascades such as the interferon response. Several viruses of clinical interest, including coronaviruses, express hydrolases that reverse ADP-ribosylation catalysed by host enzymes, suggesting an important role for this modification in host-pathogen interactions. However, which ADP-ribosyltransferases mediate host ADP-ribosylation, what proteins and pathways they target and how these modifications affect viral infection and pathogenesis is currently unclear. Here we show that host ADP-ribosyltransferase activity induced by IFNγ signalling depends on PARP14 catalytic activity and that the PARP9/DTX3L complex is required to uphold PARP14 protein levels via post-translational mechanisms. Both the PARP9/DTX3L complex and PARP14 localise to IFNγ-induced cytoplasmic inclusions containing ADP-ribosylated proteins, and both PARP14 itself and DTX3L are likely targets of PARP14 ADP-ribosylation. We provide evidence that these modifications are hydrolysed by the SARS-CoV-2 Nsp3 macrodomain, shedding light on the intricate cross-regulation between IFN-induced ADP-ribosyltransferases and the potential roles of the coronavirus macrodomain in counteracting their activity.**

**Keywords** ADP-Ribosylation; PARP; Interferon; Coronavirus; Innate Immunity
**Subject Categories** Immunology; Post-translational Modifications & Proteolysis

See also: **P Kar et al**

## Introduction

Post-translational modification by ADP-ribosylation (ADPr) is catalysed by a class of enzymes termed ADP-ribosyltransferases (ARTs), also known as poly-ADPr-polymerases (PARPs), which utilise NAD+ as a substrate to covalently attach ADP-ribose units onto amino acid sidechains or nucleic acids, in the form of either mono-ADPr (MAR) or chains of poly-ADPr (PAR) (Luscher et al,

2021). It is becoming increasingly evident that this modification plays central roles in several cellular processes, such as DNA damage signalling, chromatin remodelling, transcriptional regulation, cell death and, most relevant here, antiviral signalling (Hoch and Polo, 2019). In response to viral infections, the interferon response, which is a central component of innate immunity, induces the expression of antiviral factors collectively termed interferon-stimulated genes (ISGs), that restrict virus entry and replication and alert neighbouring cells and the adaptive immune response of the presence of a threat (Schoggins, 2019). Several ADP-ribosyltransferases, in particular PARPs 7 to 14, are ISGs and participate both in feedback regulation of the interferon response or in direct antiviral effector functions (Hoch, 2021). The importance of ADP-ribosylation in this context is highlighted by the presence of ADPr-hydrolysing macrodomains in several viruses of the alphavirus and coronavirus families, and the observation that mutation of these viral macrodomains leads to severe viral attenuation (Abraham et al, 2018; Grunewald et al, 2019). Indeed, these observations have led to large efforts to develop macrodomain inhibitors as a new class of antivirals (Gahbauer et al, 2023; Leung et al, 2022; Schuller et al, 2021). However, a comprehensive understanding of the targets and functions of host ADP-ribosylation during interferon signalling and how their reversal by viral macrodomains might benefit viral infections is currently lacking.

We have previously shown that activation of both type I or type II interferon signalling induces detectable ADP-ribosylation in human cells and that ectopic expression of the SARS-CoV-2 Nsp3 macrodomain 1 (Mac1) can reverse this modification (Russo et al, 2021). We also identified a prominent role of the PARP9/DTX3L complex in promoting IFN-induced ADP-ribosylation. PARP9 is a member of the macroPARP family, which is characterised by the presence of multiple macrodomains in addition to the PARP catalytic domain. While PARP9 seems to lack crucial amino acids required for ADP-ribosyltransferase activity, it forms a stable, constitutive complex with DTX3L, which contains a RING domain typically found in E3 ubiquitin ligases, and has been shown to catalyse the covalent linkage between ADPr and ubiquitin (Chatrin et al, 2020; Juszczynski et al, 2006; Takeyama et al, 2003; Yang et al, 2017; Zhu et al, 2022). The molecular consequences of this activity are, however, currently unclear, although PARP9 and DTX3L have been shown to regulate DNA damage signalling (Yan et al, 2013), proteasomal degradation of viral proteins (Zhang et al, 2015) and activation of the androgen receptor (Yang et al, 2021), among other functions (Vela-Rodríguez

[1]Department of Biochemistry, University of São Paulo, São Paulo 05508-000, Brazil. [2]These authors contributed equally: Victoria Chaves Ribeiro, Lilian Cristina Russo.
✉E-mail: nicolas@iq.usp.br

and Lehtiö, 2022). Interestingly, the *PARP9* and *DTX3L* genes are located in a head-to-head orientation on chromosome 3q21 and share a common bidirectional interferon-responsive promoter (Juszczynski et al, 2006). Conspicuously, the genes encoding the two other human macroPARPs, *PARP14* and *PARP15*, are located in very close proximity. While there is little evidence for the involvement of PARP15 in IFN responses (Hoch, 2021), PARP14 is an IFN-induced mono-ADP-ribosyltransferase that has been shown to participate in IFN signalling by facilitating the recruitment of RNA polymerase II to the IFN beta locus (Caprara et al, 2018). Importantly, both PARP12 and PARP14 were shown to be the main host ADP-ribosyltransferases counteracted by the coronavirus macrodomain (Grunewald et al, 2019; Kerr et al, 2023), suggesting a crucial role in antiviral immunity (Parthasarathy and Fehr, 2022). Similar to many other members of the family, PARP14 was shown to auto-modify (Vyas et al, 2013; Vyas et al, 2014) and efforts to identify PARP14 targets have been ongoing (Buch-Larsen et al, 2020; Carter-O'Connell et al, 2018; Đukić et al, 2023; Higashi et al, 2019), but a clear mechanistic understanding of its functions is still lacking. There is also evidence for crosstalk between PARP9/DTX3L and PARP14 in shaping IFN responses (Iwata et al, 2016).

Here, we show that the bulk of IFNγ-induced ADP-ribosylation depends on PARP14 catalytic activity and that PARP14 protein levels are regulated by the PARP9/DTX3L complex in concert with PARP14 itself. We find that all three of these proteins localise to cytosolic IFNγ-induced ADP-ribosylation sites and that, in addition to auto-modification, PARP14 promotes the ADP-ribosylation of DTX3L. We also show evidence that ectopic expression of the SARS-CoV-2 Nsp3 macrodomain (Mac1) may reverse both of these PARP14-mediated modifications, suggesting that PARP14 and DTX3L are cellular targets of the coronavirus macrodomain.

## Results

We have previously shown that activation of both type I or type II interferon responses induces protein ADP-ribosylation that can be detected in a punctate cytosolic pattern by immunofluorescence staining using an *Af1521*-based pan-ADPr detection reagent (MABE1016, developed by (Gibson et al, 2017)) (Russo et al, 2021). This signal was shown to be detectable in different cell lines (A549 and RPE-1), and to be reduced in RPE-1 DTX3L or PARP9 KO cells and to be sensitive to ectopic expression of the SARS-CoV-2 Nsp3 macrodomain in A549 cells (Russo et al, 2021). To further characterise the chemical nature of this modification, we used two recently developed mono-ADPr-specific antibodies (AbD33204 and AbD33205, developed by (Bonfiglio et al, 2020; Longarini et al, 2023)). Both reagents detected a similar IFNγ-induced cytoplasmic signal, which indeed extensively co-localised with an improved pan-ADPr detection reagent (eAf1521-Fc, developed by (Nowak et al, 2020)) (Fig. 1A,B). Interestingly, this

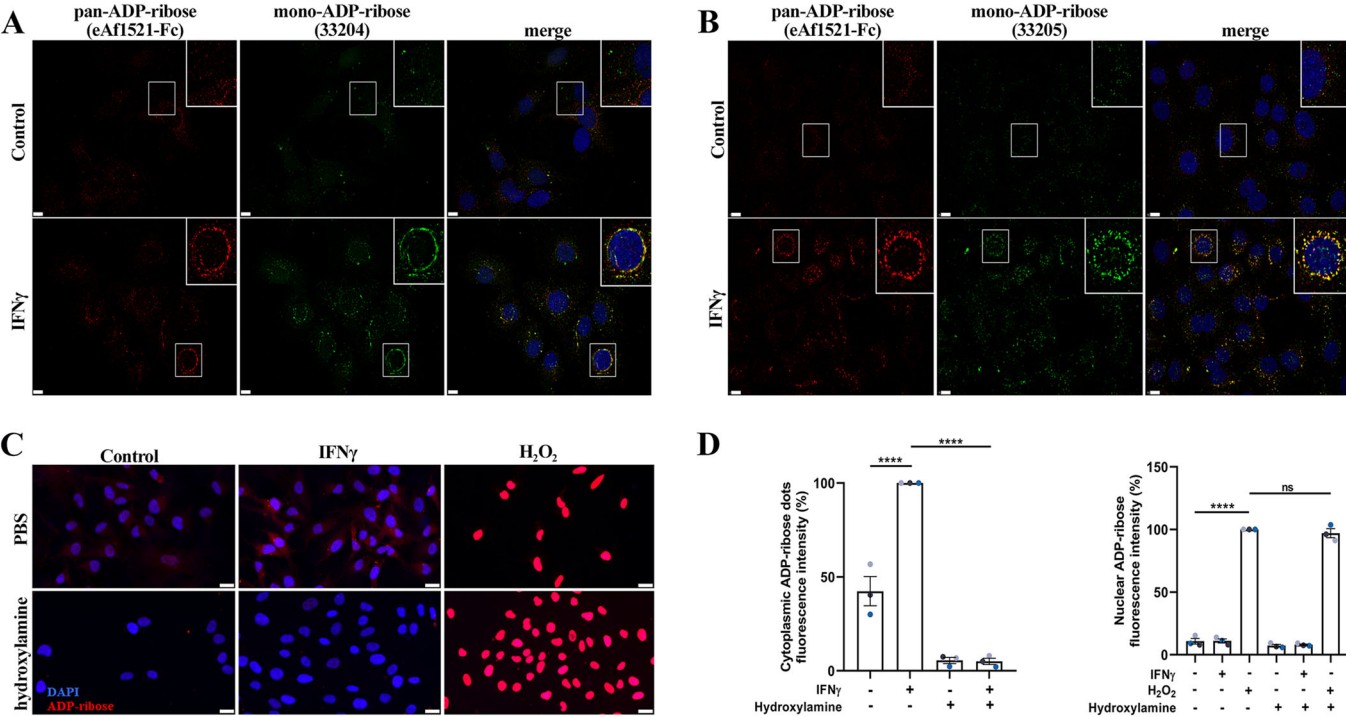

**Figure 1.  IFNγ treatment induces mono-ADP-ribosylation on acidic residues.**

(A, B) Representative immunofluorescence microscopy images of A549 cells treated or not with 500 U/mL of IFNγ for 24 h, co-stained using the indicated ADPr-specific reagents. Regions marked with a white box are enlarged in the top right corner. Scale bar: 10 μm. (C) Representative immunofluorescence microscopy images of pan-ADP-ribose (MABE1016) staining in RPE-1 cells treated with vehicle control, 100 U/mL IFNγ for 24 h or 600 μM hydrogen peroxide for 10 min. After cell fixation and permeabilisation, samples were either treated with PBS or 1 M hydroxylamine pH 7.0 for 1 h. Scale bar: 30 μm. (D) Quantification of pan-ADP-ribose signal contained in cytosolic puncta or nuclei in RPE-1 cells treated as in (C). Mean ± SEM ($n = 3$) ****$p < 0.0001$. Source data are available online for this figure.

signal was highly sensitive to hydroxylamine, which is known to hydrolyse the ester bond between ADPr units and acidic amino acid sidechains (Fig. 1C,D) (Moss et al, 1983). Demonstrating the specificity of this assay, hydroxylamine incubation did not remove PARP1-mediated nuclear ADP-ribosylation induced by hydrogen peroxide, which is thought to predominantly target serine residues and was therefore largely hydroxylamine-resistant (Fig. 1C,D) (Leidecker et al, 2016; Palazzo et al, 2018). Corroborating these results, we detected IFNγ-induced ADP-ribosylation by western blotting using a recently developed mono-ADPr-specific reagent (Abd43647, developed by (Longarini et al, 2023)), and this signal was also sensitive to hydroxylamine incubation of the lysates (Fig. EV1A). These data suggest that IFNγ treatment induces mono-ADP-ribosylation of cytoplasmic proteins on glutamate and/or aspartate residues.

Our previous findings implicated the PARP9/DTX3L complex in promoting this IFN-induced ADP-ribosylation, but did not exclude the participation of other IFN-responsive ADP-ribosyltransferases (Russo et al, 2021). To systematically evaluate this, we depleted each of the IFN-induced ADP-ribosyltransferases (PARP7 to PARP14) and DTX3L in A549 cells using siRNA (Fig. EV2A) and quantified IFNγ-induced ADP-ribosylation in these cells (Fig. 2A,B). As expected, IFNγ treatment induced robust ADP-ribosylation in control cells, and the depletion of PARP9 or DTX3L

reduced this signal (Fig. 2A,B). While the depletion of most tested enzymes had little effect, the depletion of PARP14 strongly suppressed ADP-ribosylation induced by IFNγ treatment, and also impacted the basal signal observed in untreated cells (Figs. 2A,B and EV2B). Confirming these results, we observed that siRNA depletion of PARP14 or of DTX3L also impaired the IFNγ-induced mono-ADPr signal detected on several bands by western blotting (Fig. EV2C). Under these conditions, none of the siRNA treatments affected IFNγ-induced STAT1-Y701 phosphorylation, indicating that the effects on ADP-ribosylation were not due to an impaired IFNγ signalling cascade (Fig. EV2D). To corroborate these findings, we employed the selective PARP14 inhibitor RBN012759 (Schenkel et al, 2021), and quantified ADP-ribosylation in response to either poly(I:C) or IFNγ treatment, which induce type I or type II IFN signalling respectively. In agreement with the above results, PARP14 inhibition completely prevented both poly(I:C) and IFNγ-induced ADP-ribosylation (Fig. 2C,D). These results indicate that, in addition to the previously described role of the PARP9/DTX3L complex (Russo et al, 2021), the catalytic activity of PARP14 is also required for IFN-induced ADP-ribosylation in A549 cells.

To clarify how PARP9/DTX3L and PARP14 promote IFNγ-induced ADP-ribosylation, we first determined if either the PARP9/DTX3L complex or PARP14 localise to IFNγ-induced ADP-

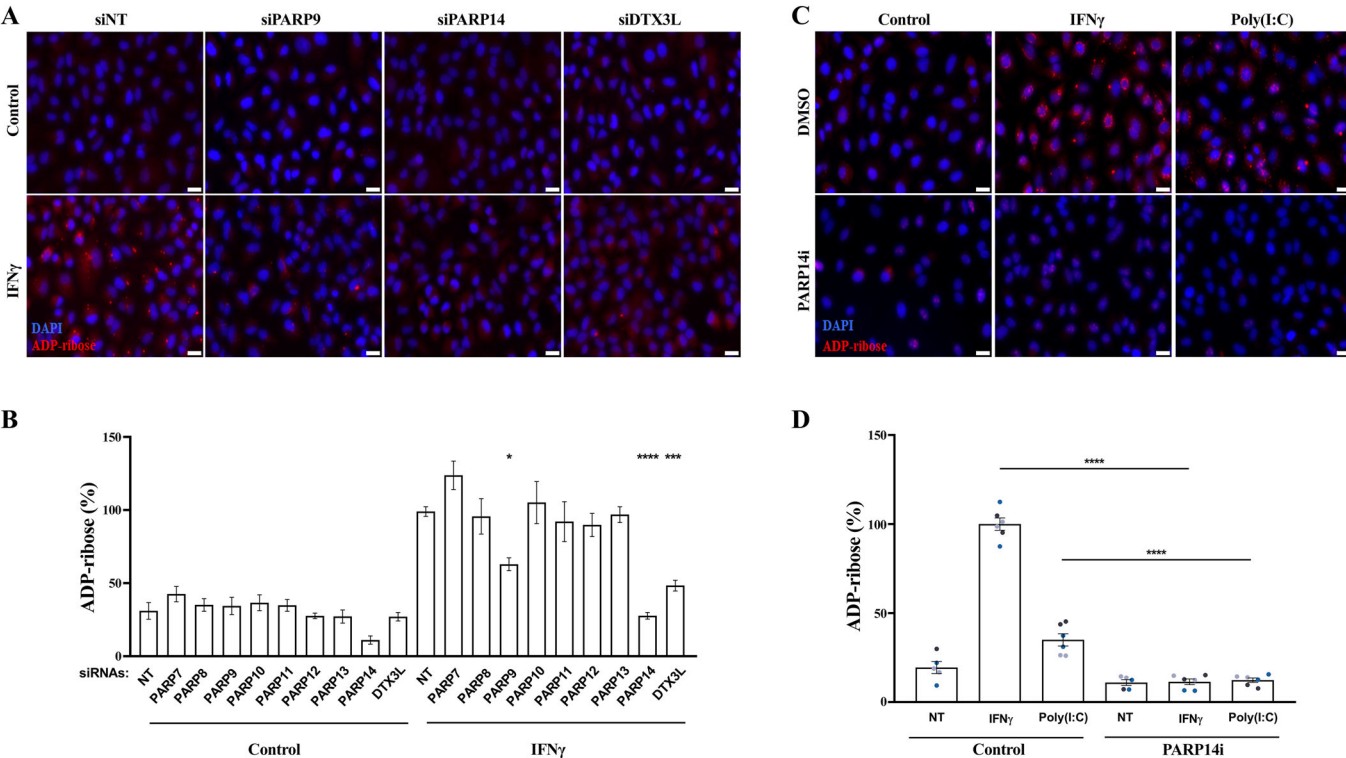

**Figure 2. PARP14 promotes IFN-induced ADP-ribosylation.**

(A) Representative immunofluorescence microscopy images and (B) quantification of pan-ADP-ribose (MABE1016) signal contained in cytosolic puncta in A549 cells transfected with indicated siRNAs, treated with vehicle control or 100 U/mL IFNγ for 24 h. (C) Representative immunofluorescence microscopy images and (D) quantification of pan-ADP-ribose (MABE1016) signal contained in cytosolic puncta in A549 cells treated with vehicle control, 100 U/mL IFNγ or transfected with 0.1 μg/mL poly(I:C) for 24 h, co-treated or not with 100 nM PARP14 inhibitor. Mean ± SEM ($n$ = 3–6, as indicated). *$p$ < 0.05, **$p$ < 0.01, ***$p$ < 0.001 and ****$p$ < 0.0001. Scale bar: 20 μm. Source data are available online for this figure.

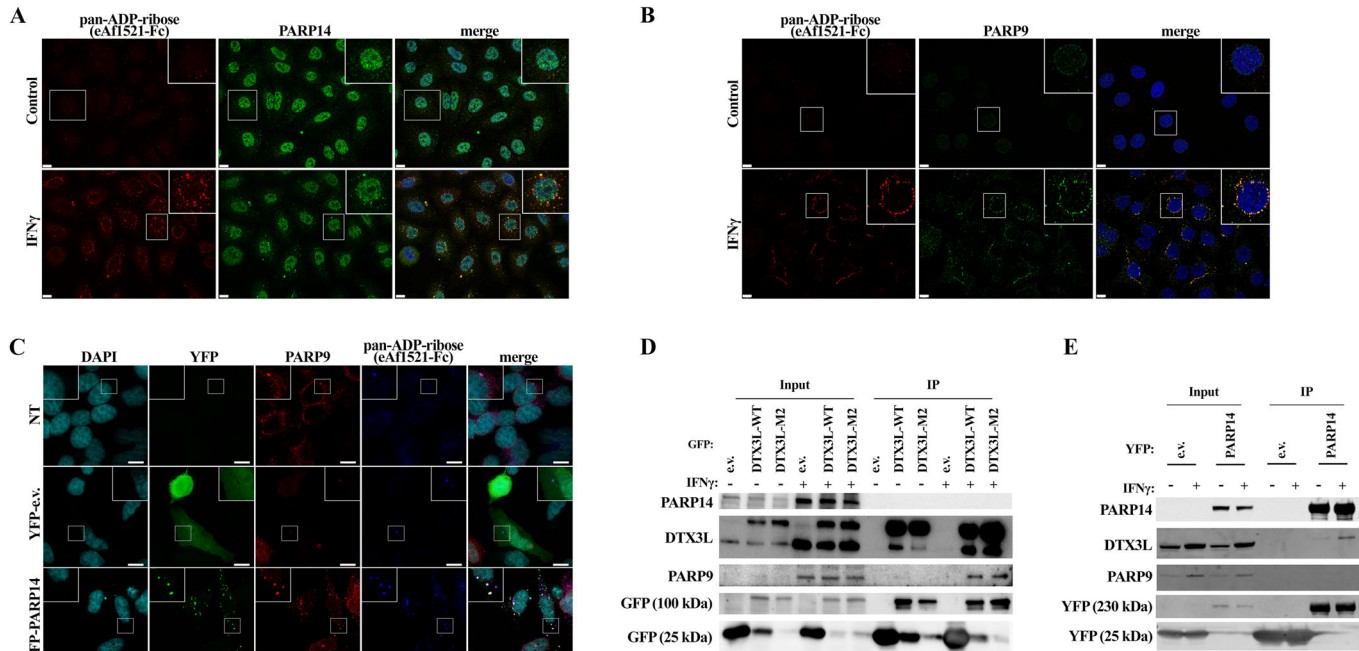

**Figure 3. PARP9, DTX3L and PARP14 co-localise with cytosolic ADP-ribosylation sites and co-precipitate.**

(A) Representative immunofluorescence microscopy images of A549 cells treated or not with 200 U/mL IFNγ for 24 h, co-stained for pan-ADP-ribose (eAf1521-Fc) and PARP14. (B) Representative immunofluorescence microscopy images of A549 cells treated or not with 500 U/mL IFNγ for 24 h, co-stained for pan-ADP-ribose (eAf1521-Fc) and PARP9. (A, B) Regions marked with a white box are enlarged in the top right corner. Scale bar: 10 μm. (C) Representative immunofluorescence confocal microscopy images of HeLa cells not transfect (NT) or transfected with YFP-empty vector (YFP-e.v.) or YFP-PARP14, treated with 200 U/mL IFNγ for 24 h, co-stained for pan-ADP-ribose (eAF1521-Fc) and PARP9. Regions marked with a white box are enlarged in the top right corner. Scale bar: 10 μm. (D) Representative immunoblot against indicated proteins in input lysates and GFP pulldown samples (IP) from HEK293FT cells transfected with empty GFP vector control (e.v.), GFP-DTX3L or GFP-DTX3L-M2 constructs, treated with vehicle control or 100 U/mL IFNγ for 24 h. (E) Representative immunoblot against indicated proteins in input lysates and GFP pulldown samples (IP) from HEK293FT cells transfected with YFP-empty vector (YFP-e.v.) or YFP-PARP14 constructs, treated with vehicle control or 200 U/mL IFNγ for 24 h. Source data are available online for this figure.

ribosylation sites. Although our PARP14 antibody detects a non-specific nuclear signal that is not sensitive to PARP14 siRNA (Fig. EV3A), the specific cytoplasmic PARP14 signal is clearly induced after IFNγ treatment and substantially co-localises with ADP-ribosylation sites both before and especially after IFNγ treatment (Figs. 3A and EV3A). Similarly, IFNγ treatment also induced the accumulation of both PARP9 and DTX3L in cytosolic inclusions that co-localise with ADP-ribosylation (Figs. 3B and EV2B,D). To corroborate these findings, we overexpressed YFP-tagged PARP14 in HeLa cells and determined its localisation relative to endogenous PARP9, DTX3L protein and ADP-ribosylation sites. In contrast to the YFP control, YFP-PARP14 extensively co-localised with both PARP9 and ADPr (Fig. 3C) or with DTX3L and ADPr (Fig. EV3E) in cytoplasmic inclusions. These data indicate that all three proteins are present within these structures. To further determine if these proteins interact with each other physically, we overexpressed GFP-DTX3L in HEK293FT cells and performed a pulldown experiment using GFP-trap beads. As expected, treatment of cells with IFNγ induced PARP9, DTX3L and PARP14 expression in input samples, indicating that HEK293FT cells are IFN-competent (Fig. 3D). Confirming the specificity of the assay, GFP-DTX3L co-immunoprecipitated PARP9, while the GFP control did not (Figs. 3D and EV3F). Interestingly, we observed that GFP-DTX3L also bound

endogenous DTX3L, suggesting that the PARP9/DTX3L complex may be composed of multiple copies of DTX3L (Fig. 3D). Importantly, we did not observe interaction between GFP-DTX3L and endogenous PARP14 in this assay (Fig. 3D). In this experiment, we also included a GFP-DTX3L-M2 mutant construct, in which four zinc-coordinating residues in the RING finger domain are mutated (C576S, H578S, C581S and C584S) (Tessadori et al, 2017), but the mutation had no effect on any of the observed interactions. To independently confirm this result, we performed the reciprocal experiment, pulling down overexpressed YFP-PARP14 using GFP-trap beads and probing for endogenous DTX3L and PARP9 in these samples. While we could not observe an interaction between YFP-PARP14 and PARP9, we could detect DTX3L in the YFP-PARP14 pulldown, indicating that these proteins interact (Figs. 3E and EV3G). Collectively, these results suggest that DTX3L can interact with both PARP9 and PARP14.

We reasoned that the requirement of both the PARP9/DTX3L complex and PARP14 for the formation of IFNγ-induced ADP-ribosylation could be explained if they operate in the same pathway, with one of these factors regulating the other. To test this hypothesis, we depleted either PARP9, DTX3L or PARP14 using siRNA and determined the protein levels of all three proteins in A549 cells. As expected, the siRNA knockdowns led to the efficient depletion of each target protein, and IFNγ treatment induced the

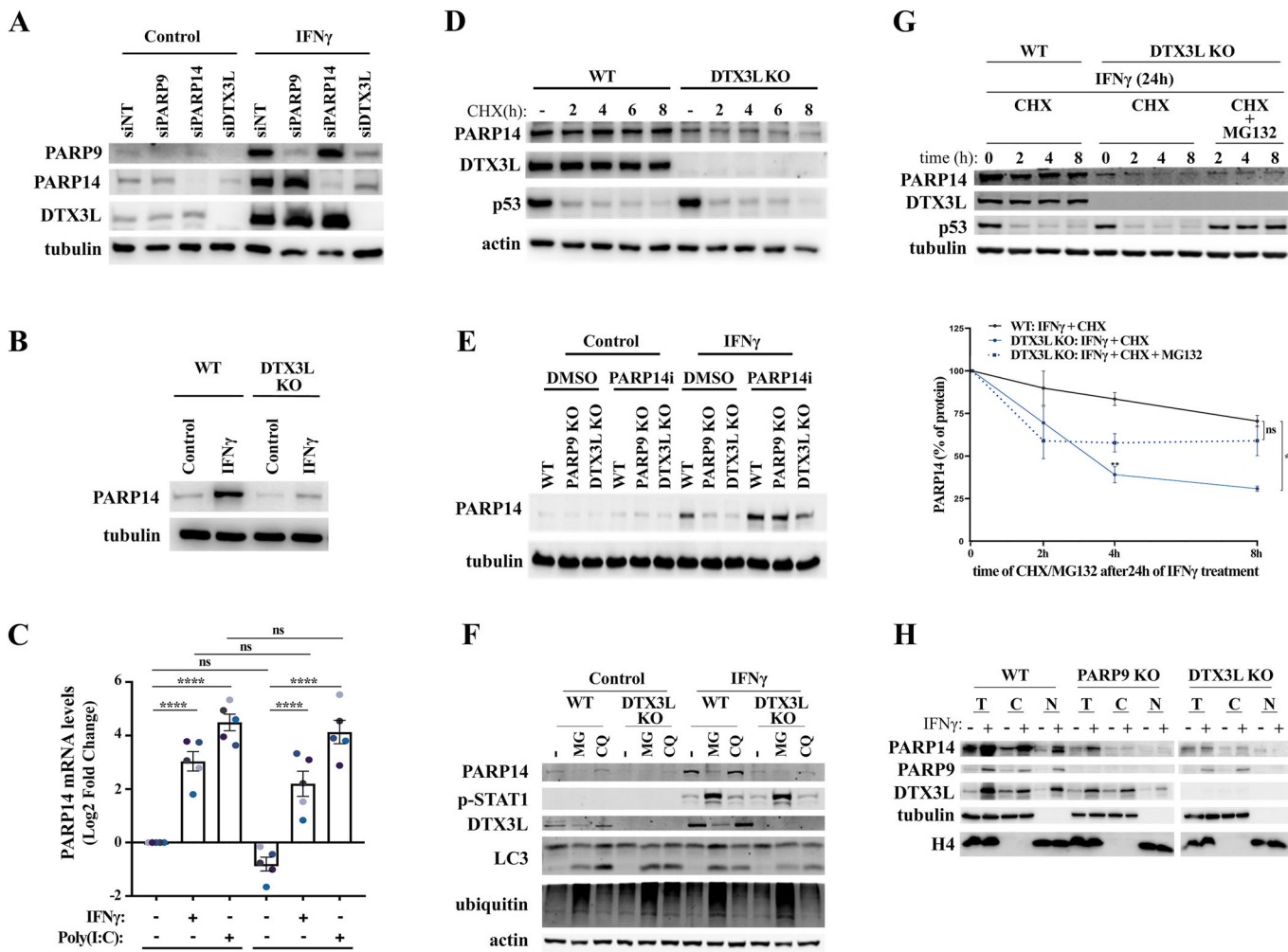

**Figure 4. The PARP9/DTX3L complex regulates PARP14 protein stability.**

(A) Representative image of immunoblots for PARP9, DTX3L and PARP14 protein levels relative to tubulin loading control in A549 cells transfected with indicated siRNAs and treated with vehicle control or 100 U/ml IFNγ for 24 h. (B) Representative image of immunoblots for PARP14 protein levels relative to tubulin loading control in RPE-1 WT or DTX3L KO cells treated with vehicle control or 100 U/mL IFNγ for 24 h. (C) Quantification of relative PARP14 mRNA levels by RT-qPCR in RPE-1 WT or DTX3L KO cells 24 h after treatment with vehicle control, 100 U/mL IFNγ or transfection with 0.1 μg/mL poly(I:C). (D) Representative image of immunoblots for indicated proteins in RPE-1 WT or DTX3L KO cells treated with 50 μg/mL cycloheximide (CHX) for indicated times. (E) Representative immunoblot for PARP14 protein levels and tubulin loading control in RPE-1 WT, PARP9 KO or DTX3L KO cells treated with vehicle controls or 100 U/mL IFNγ and/or 100 nM PARP14i for 24 h, as indicated. (F) Representative image of immunoblots for indicated proteins in RPE-1 WT or DTX3L KO cells treated with vehicle controls (-) or 100 U/mL IFNγ, 20 μM chloroquine (CQ) and/or 10 μM MG132 (MG) for 24 h, as indicated. (G) Representative image (left) and quantification (right) of immunoblots for PARP14, DTX3L and p53 protein levels and tubulin loading control in RPE-1 WT or DTX3L KO cells treated with 100 U/mL IFNγ for 24 h and subsequently treated with 50 μg/mL cycloheximide (CHX) or co-treated with 50 μg/mL cycloheximide (CHX) and 10 μM MG132 (MG) for the indicated times. (H) Representative image of immunoblots for indicated proteins in total cell lysates (T), cytoplasmic fractions (C) or nuclear fractions (N) obtained from RPE-1 WT, PARP9 KO or DTX3L KO cells treated with vehicle control or 100 U/mL IFNγ for 24 h. Mean ± SEM (n = 3–5, as indicated). *p < 0.05, **p < 0.01, ***p < 0.001 and ****p < 0.0001. Source data are available online for this figure.

levels of all three proteins in non-targeting control cells (Figs. 4A and EV4A). Consistent with the known formation of a constitutive PARP9/DTX3L heterodimer, the depletion of DTX3L severely reduced the levels of its binding partner PARP9 and a similar converse effect of siPARP9 on DTX3L levels was also observed, although the latter effect was attenuated. Crucially, while the knockdown of PARP14 marginally increased PARP9 and DTX3L levels, the depletion of DTX3L led to a striking reduction in PARP14 levels (Figs. 4A and EV4A). To corroborate this finding in a different setting, we determined PARP14 protein levels in DTX3L

knockout RPE-1 cells and again observed a severe reduction in PARP14 levels in these cells compared to controls (Figs. 4B and EV4B). To determine if this regulation occurs at the transcriptional level, we compared the levels of PARP14 mRNA in WT or DTX3L KO cells by RT-qPCR. While IFNγ and poly(I:C) treatment induced the expected increase in PARP14 mRNA levels, there was no significant difference in the amounts of PARP14 mRNA between WT and DTX3L KO cells (Fig. 4C). To ascertain if the PARP9/DTX3L complex regulates the stability of PARP14 protein, we determined PARP14 protein levels in cells treated with the

translation elongation inhibitor cycloheximide (CHX). As expected, CHX treatment led to a severe and DTX3L-independent reduction in the levels of p53 protein, which was used as a positive control due to its known short half-life (Giaccia and Kastan, 1998) (Figs. 4D and EV4C). Interestingly, both PARP14 and DTX3L levels were virtually unaltered in control cells up to 8 h of CHX treatment, indicating a relatively long half-life of these proteins. Importantly, the protein stability of PARP14 was reduced in DTX3L KO cells compared to controls (Figs. 4D and EV4C). Taken together, these data indicate that the PARP9/DTX3L complex stabilises PARP14 protein levels through post-translational mechanisms.

It was previously shown that PARP14 inhibition increases PARP14 protein levels (Schenkel et al, 2021), indicating that PARP14 catalytic activity regulates its own protein stability through unknown mechanisms. We confirmed that PARP14 inhibitor treatment leads to an increase in PARP14 levels in A549 cells and also observed a mild increase in PARP9 and DTX3L levels under these conditions (Fig. EV4D), similar to the effect seen with PARP14 depletion above (Fig. 4A). To test if the reduction in PARP14 protein levels in the absence of PARP9/DTX3L also involves PARP14 catalytic activity, we treated RPE-1 PARP9 or DTX3L KO cells with PARP14 inhibitor. Surprisingly, while PARP14 inhibitor led to a smaller than twofold increase in PARP14 levels in IFNγ-treated RPE-1 WT cells, PARP14 inhibition increased PARP14 levels by up to fivefold in IFNγ-treated PARP9 or DTX3L KO cells, practically restoring PARP14 levels back to WT levels (Figs. 4E and EV4E). To test if PARP14 protein degradation depends on the proteasome or on autophagy, we inhibited these pathways with MG132 or chloroquine, respectively, and determined their effect on PARP14 levels in WT or DTX3L KO cells. While chloroquine clearly inhibited autophagy, as determined by the accumulation of LC3II (Kabeya et al, 2003), it did not affect PARP14 levels in any of the tested conditions (Figs. 4F and EV4F). On the other hand, long-term MG132 treatment induced a surprising increase in IFNγ-induced STAT1-phosphorylation and a reduction of PARP14 and DTX3L protein levels, which we ascribe to secondary effects of proteasome inhibition and/or impaired ubiquitin turnover under these conditions (Figs. 4F and EV4F). To analyse the effects of proteasome inhibition on PARP14 protein half-life without this confounding effect, we determined PARP14 protein levels in WT or DTX3L KO cells first treated with IFNγ alone, and then incubated with either cycloheximide or cycloheximide plus MG132 for up to 8 h. Confirming the results from Fig. 4D, we observed that DTX3L knockout led to a reduction in PARP14 protein half-life also in IFNγ-treated cells (Figs. 4G and EV4G). Interestingly, treatment with MG132 under these conditions mildly delayed PARP14 protein degradation in the DTX3L KO cells (Figs. 4G and EV4G). These results suggest that in the absence of the PARP9/DTX3L complex, increased PARP14 catalytic activity drives its own degradation by a mechanism that is independent of autophagy and at least partially dependent on the proteasome.

Given that both the PARP9/DTX3L complex and PARP14 have previously been implicated in DNA repair functions in the nucleus and PARP9 and DTX3L have been shown to rely on each other for nuclear localisation (Dhoonmoon et al, 2022; Juszczynski et al, 2006; Nicolae et al, 2015; Yan et al, 2013), we decided to investigate if PARP14 localisation to the nucleus required the PARP9/DTX3L complex. Using clean biochemical fractionation conditions in which tubulin is exclusively in the cytosolic fraction and histone H4

is exclusively in the nuclear fraction, we observed that all three proteins are indeed localised both in the cytoplasm and the nucleus in RPE-1 WT cells, both before and after IFNγ treatment (Fig. 4H). In overall agreement with previous publications, the deletion of PARP9 had a mild effect on input DTX3L levels and somewhat impaired its nuclear localisation. Conversely, DTX3L KO had a more pronounced effect on PARP9 levels and completely prevented the detection of PARP9 in nuclear fractions. However, while the deletion of these factors again led to an overall reduction in PARP14 levels, this did not substantially alter its subcellular distribution (Fig. 4H).

Next, we compared the effect of IFNγ, DTX3L KO and/or PARP14 inhibitor treatment on the ADP-ribosylation status of PARP9, DTX3L and PARP14 using GST-*Af1521* pulldowns followed by western blotting (Grimaldi et al, 2018). As expected, DTX3L KO, IFNγ and PARP14 inhibitor induced the above-reported effects on PARP9, DTX3L and PARP14 protein levels in input samples (Fig. 5A). While there was only residual binding of the tested proteins to the G42E mutant GST-*Af1521* control, we detected both DTX3L and PARP14, but not PARP9, in GST-*Af1521* WT pulldowns, suggesting that these proteins are ADP-ribosylated in cells (Fig. 5A). Interestingly, *Af1521* binding of both PARP14 and DTX3L was already detectable under basal conditions, but substantially increased in response to IFNγ treatment (Fig. 5A). Crucially, co-treatment with PARP14 inhibitor almost completely reverted this IFNγ-induced *Af1521* interaction of PARP14 and DTX3L back to basal levels (Fig. 5A). In DTX3L KO cells, IFNγ-induced *Af1521* binding of PARP14 was also reduced compared to WT cells, but we interpret this to be a result of reduced PARP14 levels in DTX3L KO cells. To corroborate this result in an orthogonal assay, we used western blotting using the AbD43647 reagent (Đukić et al, 2023; Longarini et al, 2023) to detect mono-ADP-ribosylated proteins. As shown above (Figs. EV1A and EV2C), we observed an IFNγ-induced increase in ADP-ribosylation of endogenous proteins in A549 cells, which includes bands at the expected molecular weights for PARP14 and DTX3L (Fig. 5B). Using fluorescent secondary antibodies, we found that these same bands indeed are also detected by PARP14 and DTX3L-specific antibodies, and that this ADP-ribosylation signal is severely reduced upon PARP14 inhibition (Figs. 5B and EV5A). Collectively, these data suggest that PARP14 promotes both auto-ADP-ribosylation and DTX3L trans-ADP-ribosylation in response to IFNγ treatment.

We next performed the GST-*Af1521* pulldown in A549 cells ectopically expressing the SARS-CoV-2 Nsp3 macrodomain (Mac1) (Russo et al, 2021). In agreement with the above results, IFNγ treatment induced specific binding of DTX3L and PARP14 to the GST-*Af1521* WT beads in empty vector control cells, indicating that these proteins are ADP-ribosylated (Fig. 5C). Importantly, Mac1 expression considerably reduced the *Af1521* interaction of DTX3L, while the effect was milder, but still observable in several replicates, on PARP14 (Figs. 5C and EV5B). This result suggests that PARP14-dependent DTX3L ADP-ribosylation, and likely also PARP14 auto-modification, are cellular substrates of Mac1 hydrolase activity.

## Discussion

Protein ADP-ribosylation is increasingly recognised as an important player in antiviral mechanisms. Therefore, a better understanding of host ADP-ribosylation is critical to elucidate the

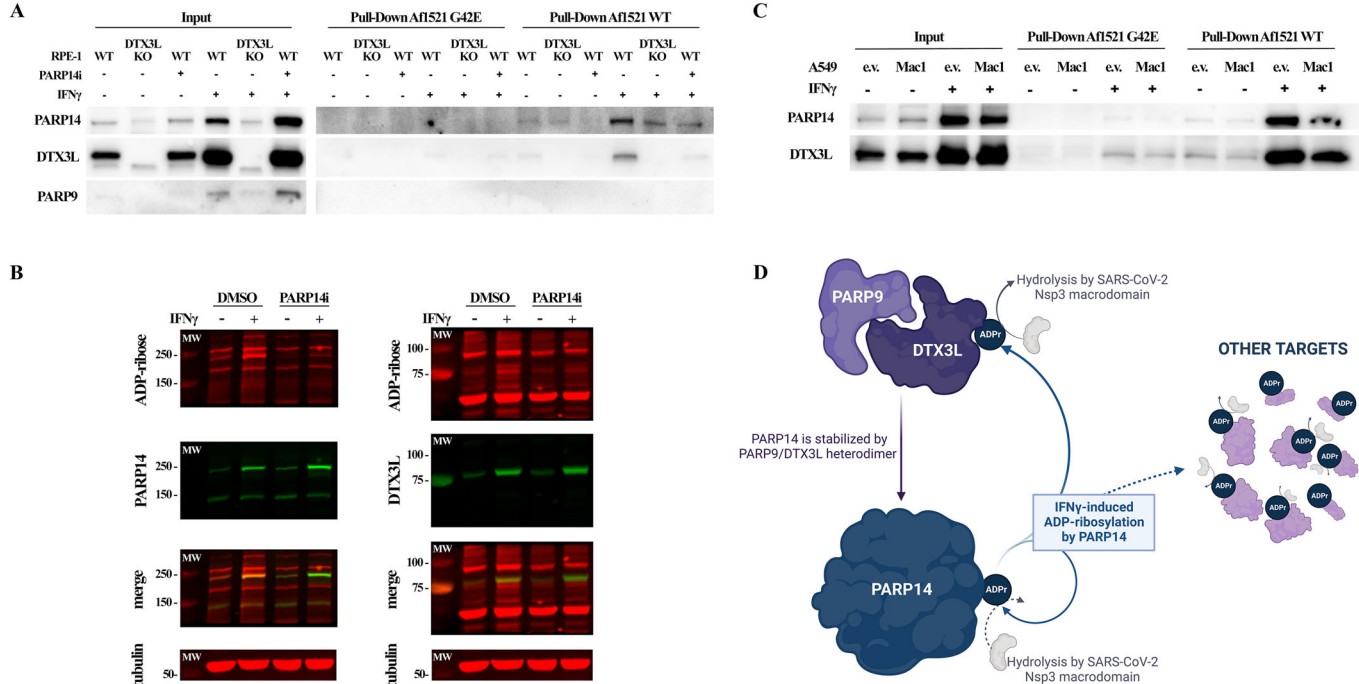

**Figure 5. PARP14 promotes auto-ADP-ribosylation and macrodomain-sensitive DTX3L trans-ADP-ribosylation.**

(A) Representative immunoblots for PARP14, DTX3L and PARP9 in whole cell lysates (input) and in material bound to GST-*Af1521-G42E* or GST-*Af1521*-WT beads from RPE-1 WT or DTX3L KO cells treated with 100 U/mL IFNγ and/or 100 nM PARP14 inhibitor, as indicated. (B) Representative fluorescent immunoblots for ADP-ribosylated proteins (red) and either PARP14 (left) or DTX3L (right) (both in green) in samples from A549 cells treated with vehicle control or 100 U/mL IFNγ for 24 h and/or 100 nM PARP14 inhibitor, as indicated. (C) Representative immunoblots for PARP14 and DTX3L in whole cell lysates (input) and in material bound to GST-*Af1521-G42E* or GST-*Af1521*-WT beads from A549 cells transduced with either empty vector (e.v.) or SARS-CoV-2 Nsp3 macrodomain 1 (Mac1) lentiviral constructs, treated with vehicle control or 100 U/mL IFNγ for 24 h, as indicated. (D) Schematic representation of the proposed model. The PARP9/DTX3L complex regulates PARP14 protein levels and PARP14 catalyses ADP-ribosylation of itself, DTX3L and likely other targets. The SARS-CoV-2 Nsp3 macrodomain can hydrolyse this modification on PARP14 and DTX3L. Created with BioRender.com. Source data are available online for this figure.

molecular functions of viral macrodomains such as the SARS-CoV-2 Nsp3 Mac1, which is a promising therapeutic target (Hoch, 2021; Leung et al, 2022). Here, we used several different reagents and methodologies that allow the detection of protein ADP-ribosylation in response to interferon-gamma signalling, and studied the role of the PARP9/DTX3L complex and PARP14 in promoting this modification.

We show that IFNγ-induced ADP-ribosylation can be detected by immunofluorescence staining using two pan-ADPr detection reagents and two mono-ADPr-specific reagents, and by western blotting using one mono-ADPr-specific reagent, indicating that at least part of the modification is mono-ADPr. Although the mono-ADPr detection reagents were raised against serine-modified targets, they can also detect mono-ADP-ribosylation on other target residues (Bonfiglio et al, 2020; Longarini et al, 2023), precluding a clear identification of amino acid linkages using these reagents alone. However, both the immunofluorescence and the western blotting signals were sensitive to hydroxylamine treatment, indicating a linkage via acidic amino acids (glutamate/aspartate). Given that glutamate/aspartate linkages are known to be labile (Tashiro et al, 2023; Weixler et al, 2023), this could also explain why IFNγ-induced ADP-ribosylation is more reliably detected by immunofluorescence staining than by western blotting. We propose that PARP14 is the main host ADP-ribosyltransferase that catalyses

this IFNγ-induced ADP-ribosylation in epithelial cells, as PARP14 depletion or treatment with a specific PARP14 inhibitor completely prevented this modification and depletion of other candidate ADP-ribosyltransferases had no clear effect on IFNγ-induced ADP-ribosylation. These data are in agreement with evidence that PARP14 is a mono-ADP-ribosyltransferase that predominantly targets acidic residues (Carter-O'Connell et al, 2018; Higashi et al, 2019; Javed et al, 2023; Tashiro et al, 2023; Wallace et al, 2021).

We also reconcile our previous observations that the PARP9/DTX3L complex was required for IFN-induced ADP-ribosylation (Russo et al, 2021) with the above findings indicating an important role for PARP14 in this process. We show here that both are likely to be true, and that, in fact, the PARP9/DTX3L complex contributes to IFNγ-induced ADP-ribosylation indirectly, via regulation of PARP14 protein levels. Interestingly, our data indicate that this regulation is unlikely to be mediated through transcriptional control of PARP14 mRNA levels or control of PARP14 degradation by autophagy, but that proteasome activity is at least partially responsible for the reduced PARP14 protein stability in the absence of PARP9/DTX3L (Fig. 4). During the revision of this manuscript, a similar effect of PARP9 or DTX3L depletion on PARP14 protein stability, but not on PARP14 mRNA levels, was reported by the Grundy group (Saleh et al, 2024). While it remains to be determined exactly how the PARP9/DTX3L complex

regulates PARP14 degradation, our data suggest that PARP14 catalytic activity is a crucial player in this mechanism, as PARP14 inhibition restored PARP14 protein levels in PARP9 or DTX3L KO cells. One possible interpretation of this data is that the PARP9/DTX3L complex inhibits PARP14 activity, and that unrestrained PARP14 activity induces its own turnover, perhaps in a proteasome-dependent mechanism. Evidence in favour of a role for PARP9/DTX3L as negative regulators of PARP14 catalytic activity is presented in the accompanying manuscript by the Ahel group (Kar et al, 2024).

Cross-regulation between PARP9/DTX3L and PARP14 has already been observed in macrophages (Iwata et al, 2016). However, PARP9/DTX3L and PARP14 were proposed to have antagonistic functions, and were suggested to be involved in the regulation of the IFN response itself. Both here and in our previous study (Russo et al, 2021), we did not observe an effect of PARP9/DTX3L or PARP14 depletion on IFNγ-induced STAT1-phosphorylation, indicating that the roles of PARP9/DTX3L and PARP14 are substantially different in macrophages as compared to the epithelial cells used in our studies. Considering that the effects and consequences of IFN signalling are considerably different between immune cells and epithelial cells, this is perhaps not unexpected.

Regarding the PARP9/DTX3L complex itself, we observed that the nuclear localisation of PARP9 depends on DTX3L expression, whereas the cytosolic pool of PARP9 is less affected by DTX3L loss. This is reminiscent of previous observations suggesting that these proteins are shuttled in and out of the nucleus, and rely on each other for adequate localisation (Juszczynski et al, 2006). These observations may have a bearing on the nuclear functions of the PARP9/DTX3L heterodimer in DNA repair (Yan et al, 2013) and regulation of androgen receptor signalling (Yang et al, 2021). We also provide circumstantial evidence that the PARP9/DTX3L complex may be composed of an oligomeric assembly of these proteins, as GFP-DTX3L interacted with endogenous DTX3L. This is in agreement with recent biochemical evidence in favour of oligomerisation of these proteins (Ashok et al, 2022; Saleh et al, 2024). We further present evidence that PARP14 interacts with DTX3L, under conditions in which we could not detect an interaction with PARP9. However, there is some evidence in the literature that these proteins form a ternary complex (Bachmann et al, 2014; Caprara et al, 2018; Iwata et al, 2016; Saleh et al, 2024), so whether this difference is caused by technical differences in co-IP procedures or a biologically significant difference between cell lines, remains to be determined. Our data also indicate that all three of these proteins can localise both to the nucleus and to the cytoplasm, which is again consistent with previously proposed functions of PARP14 in DNA repair (Dhoonmoon et al, 2022; Nicolae et al, 2015) and in the regulation of RNA polymerase II recruitment to chromatin (Caprara et al, 2018).

Our data indicate that upon IFNγ treatment, the cytosolic fraction of PARP9, DTX3L and PARP14 localises to punctate inclusions also containing IFNγ-induced ADP-ribosylation. The formation of these inclusions can be observed under several different conditions (e.g. using either PFA or methanol fixation, endogenous or epitope-tagged proteins and different ADPr-detection reagents and cell lines), but the full composition of this structure and its functions during the interferon response remain to be clarified.

We provide evidence that both PARP14 itself and DTX3L are targets of PARP14 mono-ADPr during the interferon response. This is consistent with evidence that PARP14 can auto-modify both in cells and in vitro (Carter-O'Connell et al, 2018; Đukić et al, 2023; Rack et al, 2020; Vyas et al, 2013; Vyas et al, 2014), including the detection of PARP14 ADP-ribosylation in IFNγ-treated macrophages (Higashi et al, 2019), and with data in the accompanying manuscript showing DTX3L trans-ADP-ribosylation by PARP14 in vitro (Kar et al, 2024). Interestingly, we observed that ectopic expression of the isolated SARS-CoV-2 Mac1 can reduce the binding of both PARP14 and DTX3L to Af1521 beads, indicating that the coronavirus macrodomain can hydrolyse the ADP-ribose modification of these proteins. However, we cannot rule out that the presence of the viral macrodomain in the pulldown lysates interfered with Af1521 binding to ADP-ribosylated proteins by competition. Nonetheless, this result would be consistent with our previous data showing that IFN-induced ADP-ribosylation, which we show here relies heavily on PARP14 catalytic activity, is sensitive to Mac1 overexpression (Russo et al, 2021), as well as data showing that PARP14 auto-modification can be reversed by the SARS-CoV-2 macrodomain in vitro (Đukić et al, 2023; Rack et al, 2020), and that expression of this domain can hydrolyse ADPr modifications induced by PARP14 overexpression (Đukić et al, 2023). This finding is also in agreement with the observation that PARP14 preferentially catalyses ADPr modification on acidic amino acids (Carter-O'Connell et al, 2018; Higashi et al, 2019; Javed et al, 2023; Tashiro et al, 2023; Wallace et al, 2021), which indeed are the preferential target residues of hydrolytic macrodomains (Đukić et al, 2023; Jankevicius et al, 2013; McPherson et al, 2017; Rosenthal et al, 2013). Interestingly, the PARP14 macrodomain 1, which was recently shown to be catalytically active (Đukić et al, 2023; Torretta et al, 2023), is a potential evolutionary ancestor of the coronavirus macrodomain, which was likely acquired by these viruses through horizontal gene transfer from a host macroPARP, of which PARP14 is thought to be the most ancient (Delgado-Rodriguez et al, 2023; Rack et al, 2020). While the functions of putative PARP14 and DTX3L modifications remain to be determined, their reversal by the coronavirus macrodomain suggests that they might play important antiviral functions. However, given that ectopic overexpression of hydrolytic macrodomains is often observed to reduce ADP-ribosylation of cellular proteins with an apparently low specificity (Delgado-Rodriguez et al, 2023; Đukić et al, 2023), further studies in which the coronavirus macrodomain functions in its native environment are required to confirm these observations. Nonetheless, PARP14 was identified as one of the fastest-evolving genes in the primate lineage (Daugherty et al, 2014), and both PARP12 and PARP14 were found to be the main host ADP-ribosyltransferases countering the replication and pathogenesis of coronaviruses lacking a functional macrodomain (Grunewald et al, 2019; Kerr et al, 2023).

In conclusion, we propose a model in which PARP14 is the main IFNγ-induced ADP-ribosyltransferase, whose protein stability is regulated by the PARP9/DTX3L complex. We further propose that PARP14 modifies both itself and DTX3L and that these modifications can be hydrolysed by the SARS-CoV-2 Nsp3 macrodomain 1 (Fig. 5D). These findings have important implications for our understanding of the role of ADP-ribosylation in antiviral

signalling responses and in elucidating the functions of viral macrodomains, which are novel targets for antiviral therapeutics.

## Methods

### Cell culture

All cell lines were maintained at 37 °C in a humidified atmosphere containing 5% CO$_2$. A549 lung adenocarcinoma (ATCC: CRM-CCL-185) and HeLa (ATCC: CCL-2) cells were grown in DMEM/high glucose media (Thermo) and RPE-1 hTERT (ATCC: CRL-4000) retinal pigment epithelial cell lines were grown in DMEM/F-12 (Thermo) supplemented with 15 mM HEPES (Merck). Both media were also supplemented with inactivated 10% foetal bovine serum (FBS—Thermo) and 1% Penicillin-Streptomycin 10,000 U/mL (Thermo). HEK293FT (Thermo: R70007) cells were grown in DMEM/high glucose media (Thermo) supplemented with non-essential amino acids (NEAA), L-glutamine, sodium pyruvate, gentamicin sulphate and 10% FBS. The CRISPR/Cas9-generated RPE-1 PARP9 KO and DTX3L KO cells, as well as the lentivirus-transduced A549 cells expressing SARS-CoV-2 Mac1 have been previously generated (Russo et al, 2021). Cells were routinely tested for mycoplasma contamination by PCR.

### Treatment conditions

Recombinant interferon γ (Sigma SRP3058) was used at the indicated doses (100 to 500 U/mL), the PARP14 inhibitor RBN012759 (TargetMol) was used at 100 nM and poly(I:C) (Pharmacia, ref 27473201) was transfected using PEI (Thermo) to a final concentration of 0.1 µg/mL. All treatments were performed for 24 h. Cycloheximide (Sigma) was used at a final concentration of 50 µg/mL, chloroquine (Sigma) at 20 µM and MG132 (Sigma) at 10 µM for indicated times. Hydrogen peroxide (Sigma) treatment was performed for 10 min at a final concentration of 600 µM in PBS.

### siRNA transfection

All indicated siRNAs were ON-TARGETplus siRNA SMARTpools (Dharmacon). Briefly, 2 pmol siRNA was mixed with 0.3 µL Lipofectamine RNAiMAX Transfection Reagent (Thermo) in reduced-serum Opti-MEM medium (Gibco) and incubated for 20 min at room temperature to allow complex formation. For reverse transfection, $0.75 \times 10^4$ trypsinised cells were added in suspension to the siRNA solution and then seeded in a 96-well plate. 24 h later, forward transfection was performed by adding a new siRNA solution prepared as above to the wells. Experiments were performed 24 h after the second transfection. For experiments using a higher number of cells, the same proportion of siRNA and lipofectamine was maintained. The siRNA efficiency was assessed by RT-qPCR or western blot. Non-targeting siRNA was included as a negative control.

### Plasmids and transfection

Full-length DTX3L and DTX3L-M2 mutant were subcloned from a pCS2+ vector backbone, a kind gift from Gijs van Haaften, UMC Utrecht, Netherlands (Tessadori et al, 2017), into a peGFP-C1 vector using EcoRI sites. A spurious R671G mutation traced back to the original constructs was corrected by site-directed mutagenesis in both WT and M2 vectors. All final clones were confirmed by Sanger sequencing of the whole insert. These plasmids were transfected into HEK293FT cells using PEI (Thermo).

Both the YFP-empty vector (YFP-e.v.) and YFP-PARP14 full length were kindly donated by I. Ahel (Đukić et al, 2023). For the GFP-trap experiments, these plasmids were transfected into HEK293FT cells using PEI (Thermo), and for the immunofluorescence experiments, these plasmids were transfected into HeLa cells using Lipofectamine™ 3000 Transfection Reagent (Thermo).

### Western blotting

Adherent cells were washed in PBS and lysed directly in preheated Laemmli buffer devoid of bromophenol blue and beta-mercaptoethanol. Lysates were transferred to tubes and boiled for 15 min at 100 °C. The protein concentration was determined using the BCA protein quantification kit (Pierce), adjusted accordingly and boiled again in the presence of bromophenol blue and beta-mercaptoethanol. 20 to 40 µg of protein per sample were loaded into standard SDS-PAGE gels, transferred to nitrocellulose membranes (Bio-Rad) and visualised with Ponceau S (Sigma). To allow the use of multiple antibodies, some membranes were cut horizontally, and the different portions of the same membrane were incubated with appropriate antibodies. Membranes were then blocked with 5% skimmed milk or 5% bovine serum albumin (BSA) for 1 h at room temperature. The respective primary antibodies were incubated overnight at 4 °C (or 30 min at room temperature for actin or tubulin loading control) with slow agitation. Following extensive washing, the membranes were incubated with the appropriate HRP-conjugated secondary antibody (Sigma) for 1 h. After a brief incubation with ECL Prime (Amersham), the signal was detected using a ChemiDoc MP Imaging System (Bio-Rad), and the signals were quantified using ImageJ software.

For mono-ADPr detection, cells were washed on ice with chilled PBS three times and lysed with RIPA lysis buffer supplemented with HALT protease inhibitor (Thermo), followed by scraping. The lysate was incubated for 20 min on a rotating wheel at 4 °C and the supernatant was collected after centrifugation at $15,000 \times g$ for 15 min at 4 °C. After adding the appropriate amount of Laemmli buffer to 25–40 µg of protein, the samples were boiled for 10 min at 70 °C. Western blot was then performed essentially as described above.

For fluorescent immunoblots, proteins were transferred onto an Immobilon-NC membrane (Millipore), the secondary antibodies were conjugated to AlexaFluor680 or AlexaFluor800 (Thermo) and the signal was detected using a ChemiDoc MP Imaging System (Bio-Rad).

### Immunofluorescence staining

Cells were seeded on 1.5H glass coverslips (Thorlabs), treated as indicated, washed with PBS, and fixed with 4% EM-grade PFA (EMS), with subsequent quenching with 0.1 M glycine. After permeabilisation in 0.2% TritonX-100 in PBS, samples were

blocked in 10% inactivated foetal bovine serum diluted in PBS (or diluted in 0.2% TritonX-100 for ADP-ribose) for 1 h and incubated with primary antibody for 1 h at room temperature or overnight at 4 °C in a humid chamber. Samples were washed, incubated with appropriate fluorescently labelled secondary antibodies (Thermo) at 1:1000 dilution for 1 h at room temperature, washed again and stained with DAPI (Thermo). The coverslips were mounted in Vectashield (Vector Labs) and sealed with nail polish. Hydroxylamine (Sigma) treatment was performed at 1 M pH 7.0 for 1 h at room temperature, after cells were fixed and permeabilised.

## Image acquisition and analysis

For quantitative fluorescence microscopy analyses, images were acquired on a customised TissueFAXS i-Fluo system (TissueGnostics) mounted on a Zeiss Axio Observer 7 microscope (Zeiss), using 20 × Plan-Neofluar (NA 0.5) objective and an ORCA Flash 4.0 v3 camera (Hamamatsu). Images were acquired using automated autofocus settings and were analysed using StrataQuest software (TissueGnostics) following the workflow described in (Russo et al, 2021).

To assess co-localisation, images were acquired on a DMi8 widefield microscope (Leica), using a 63x objective (NA = 1.4) and submitted to blind deconvolution with the Leica Application Suite-X software (Leica Microsystems) or SP8 STED FALCON confocal microscope (Leica), using a 100x objective (NA = 1.4).

## RT-qPCR

Total RNA was extracted from $10^5$ cells after the indicated treatments using an RNeasy kit (Qiagen), following the manufacturer's protocol. After quantification using NanoDrop 1000 spectrophotometer (Thermo), the samples were treated with TURBO DNase (Ambion) and reverse-transcribed using SuperScript II RT with both oligo(dT) and random hexamer primers (Thermo), according to manufacturer's instructions. qPCR was then performed using Power SYBR green Master mix (Thermo) with 200 nM of the primer sets indicated below and 5 ng of cDNA per reaction. For each biological replicate there were three technical replicates. Reactions were performed on an Applied Biosystems 7500 Real-time PCR system, using default settings. RPL19 was used as housekeeping gene control and a standard $2^{-\Delta\Delta Ct}$ analysis was performed relative to the untreated control.

## Immunoprecipitation (IP)

For GFP-trap immunoprecipitation, the GFP-Trap Agarose kit (ChromoTek) was used, following the manufacturer's instructions. Briefly, HEK293FT cells were transfected with empty vector GFP, GFP-DTX3L WT or GFP-DTX3L-M2 mutant constructs or empty vector YFP or YFP-PARP14. After treatment, cells were lysed (10 mM Tris pH 7.5, 150 mM NaCl, 0.5 mM EDTA, 0.5% NP40, 1 mM PMSF, protease inhibitor cocktail), centrifuged and the protein concentrations quantified. About 200 μL of the supernatant (adjusted to the same concentration) was added to 300 μL of dilution buffer (lysis buffer without NP40) with GFP-agarose beads under slow agitation for 1 h. After washing three times with washing buffer (10 mM Tris pH 7.5, 150 mM NaCl, 0.5 mM EDTA

and 0.05% NP40), Laemmli buffer was added and the beads were boiled at 100 °C for 15 min and analysed by western blot. The whole procedure was performed on ice or at 4 °C.

## Subcellular fractionation

Cells were trypsinised and collected in PBS, followed by centrifugation at $200 \times g$ for 5 min at 4 °C. The supernatant was fully removed, and the pellet resuspended in NIB buffer (15 mM Tris pH 7.5, 60 mM KCl, 15 mM NaCl, 5 mM MgCl2, 1 mM CaCl2, 250 mM sucrose, 1 mM DTT, protease inhibitor cocktail) plus 0.3% NP40, when the total fraction was collected. After incubation on ice for 5 min followed by centrifugation at 4 °C, the supernatant containing the cytoplasmic fraction was collected. The samples were washed five times in NIB buffer devoid of detergent and centrifuged in $800 \times g$ for 5 min at 4 °C. The remaining pellet was collected as a nuclear fraction and all samples were boiled at 100 °C for 10 min in Laemmli buffer and analysed by western blot.

## Af1521-based pull-down

To purify ADP-ribosylated proteins from total cell lysates, we used an affinity-based pull-down based on the ADPr-binding macrodomain of the *Af1521* protein from *Archaeoglobus fulgidus*, as described in (Grimaldi et al, 2018). Briefly, after the indicated cell treatments, the growth medium was removed and the dishes were washed with pre-cooled PBS three times on ice. The cells were then lysed with 500 μL RIPA lysis buffer supplemented with HALT protease inhibitor (Thermo) per 150 mm dish, followed by scraping. The cell lysate was incubated for 20 min on a rotating wheel and the supernatant was collected after centrifuging at $15,000 \times g$ for 15 min. About 1–2 mg of cell lysate was incubated with 250 μL of GST-*Af1521*-G42E immobilised on glutathione sepharose beads for 8 h on a rotating wheel. *Af1521*-G42E mutant was used as a pre-clearing control because it is unable to bind ADP-ribosylated proteins. The mixture was centrifuged at $500 \times g$ for 5 min and the supernatant containing the unbound proteins was incubated with the GST-*Af1521* WT beads overnight on a rotating wheel. Both beads were washed eight times with RIPA lysis buffer. After the last wash, Laemmli buffer was added and the beads were boiled for 10 min at 95 °C. Finally, the eluate was analysed by western blotting. The procedure was performed on ice or at 4 °C.

## Statistical analyses

All experiments were performed independently at least three times, sometimes with technical replicates. All graphs and statistical analyses were generated using GraphPad Prism 8.0.1 software, here displayed with the individual data points and their mean ± SEM. The analysis was made using the normalised values relative to the IFNγ-treated control for each replicate, considered as 100%, or normalised relative to the untreated control, considered as 1x. Statistical comparisons between samples were performed using ANOVA, with $p < 0.0001$ indicated by ****, $p < 0.001$ indicated by ***, $p < 0.01$ indicated by **, and $p < 0.05$ indicated by *.

## Primers and antibodies

| Oligo name | Sequence |
|---|---|
| RPL19-Fwd | GATCGATCGCCACATGTATCAC |
| RPL19-Rev | TTGTCTGCCTTCAGCTTGTG |
| PARP14-Fwd | CTGTGTTCCCATACTATGCCTCA |
| PARP14-Rev | ACGCCTCATTTCATCGTTTATCT |
| PARP7_fwd | AATTTGACCAACTACGAAGGCTG |
| PARP7_rev | CAGACTCGGGATACTCTCTCC |
| PARP8_fwd | GGGATGTGTTCAAGGCAAGAG |
| PARP8_rev | CCGCCAACGTAGGTAAAAGTAA |
| PARP10_fwd | AGGCGGCTGAGGAGTTTCT |
| PARP10_rev | GGCGCTCTGTCCCAAAGAC |
| PARP11_fwd | AGACGATGGATCGCAACCG |
| PARP11_rev | ATGCAGATTGCTTCCACAAATTC |
| PARP12_fwd | GCCATGACTTACGGTGCTACC |
| PARP12_rev | CCAAACTCATCACTCCAGTACCA |
| PARP13_fwd | CCGGTGCAACTATTCGCAGT |
| PARP13_rev | TCAGTCCAGAGAGTTCGTGATTT |

| Antibody | Host species | Supplier (Catalogue number) |
|---|---|---|
| Actin | Mouse | Millipore (MAB1501) |
| pan-ADP-ribose | Rabbit | Millipore (MABE1016) |
| pan-ADP-ribose (eAf1521-Fc) | Mouse | A kind gift from M. Hottiger |
| Mono-ADP-Ribose (33204) | Human/Rabbit IgG chimera | Bio-Rad (HCA354) |
| Mono-ADP-Ribose (33205) | Human/Rabbit IgG chimera | Bio-Rad (HCA355) |
| Mono-ADP-Ribose (43647) | Fused to Mouse Fc | A kind gift from I. Matic |
| Mono-ADP-Ribose (43647) | Fused to HRP | A kind gift from I. Matic |
| p53 (B20.1) | Mouse | Santa Cruz Biotechnology (sc-56180) |
| STAT1 | Rabbit | Proteintech (10144-2-AP) |
| STAT1 phospho-Y701 | Rabbit | Cell Signalling (9167) |
| PARP9 | Rabbit | Thermo (40-4400) |
| PARP14 | Rabbit | Sigma (HPA012063) |
| DTX3L | Rabbit | Bethyl (A300-834A) |
| FLAG | Mouse | Sigma (F1804) |
| GFP | Rabbit | Abcam (ab290) |
| Ubiquitin (P37) | Rabbit | Cell Signalling (58395 S) |
| LC3B | Rabbit | Cell Signalling (2775 S) |
| Alpha Tubulin [DM1A] | Mouse | Abcam (ab7291) |
| Anti-Rabbit-HRP | Donkey | Sigma (SAB3700934) |
| Anti-Mouse-HRP | Donkey | Sigma (SAB3701105) |

| Antibody | Host species | Supplier (Catalogue number) |
|---|---|---|
| Anti-Mouse-AF488 | Donkey | Thermo (A21202) |
| Anti-Rabbit-AF488 | Donkey | Thermo (A21206) |
| Anti-Mouse-AF568 | Donkey | Thermo (A10037) |
| Anti-Rabbit-AF568 | Donkey | Thermo (A10042) |
| Anti-Mouse-AF680 | Goat | Thermo (A21057) |
| Anti-Rabbit-AF800 | Goat | Thermo (A32735) |

## Data availability

Raw microscopy images for the high-content imaging experiments have been deposited in the BioImage Archive (accession ID: S-BIAD1068). https://doi.org/10.6019/S-BIAD1086.

The source data of this paper are collected in the following database record: biostudies:S-SCDT-10_1038-S44318-024-00125-1.

## Peer review information

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

## Acknowledgements

We would like to thank Célia Ludio Braga for technical assistance in the NCH lab, Giovanna Grimaldi (IBBC, Naples) for the GST-*Af1521* constructs, Michael Hottiger (Univ. of Zurich) for the e*Af1521*-Fc reagent, Ivan Matic (Max Planck Institute for Biology of Ageing, Cologne) for the mono-ADPr-binding reagents, Ivan Ahel (Univ. of Oxford) for the YFP-PARP14 constructs and Gijs van Haaften (UMC, Utrecht) for the DTX3L constructs. This work was funded by FAPESP grants 2018/18007-5 and 2020/05317-6 to NCH, and FAPESP stipends 2019/25914-1 and 2023/15157-4 to VCR, and 2020/11162-5 and 2022/10947-4 to LCR. The microscopes used in this study were funded by FAPESP grants 2019/06039-2 (to NCH) and 2015/02654-3 (to Alexandre Bruni-Cardoso).

## Author contributions

**Victoria Chaves Ribeiro**: Conceptualization; Formal analysis; Validation; Investigation; Visualisation; Methodology; Writing—review and editing. **Lilian Cristina Russo**: Conceptualization; Formal analysis; Validation; Investigation; Visualisation; Methodology; Writing—review and editing. **Nicolas Carlos Hoch**: Conceptualization; Supervision; Funding acquisition; Writing—original draft; Project administration; Writing—review and editing.

Source data underlying figure panels in this paper may have individual authorship assigned. Where available, figure panel/source data authorship is listed in the following database record: biostudies:S-SCDT-10_1038-S44318-024-00125-1.

## Disclosure and competing interests statement

The authors declare no competing interests.

# Expanded View Figures

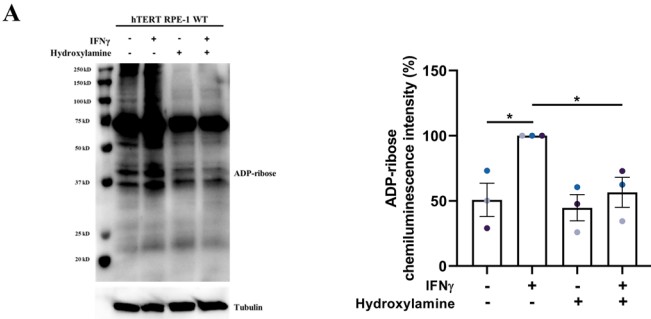

**Figure EV1.   IFNγ-induced ADP-ribosylation is hydroxylamine-sensitive.**

(related to figure 1). **(A)** Representative image (left) and quantification (right) of immunoblot analyses for mono-ADP-ribose (43647 HRP-coupled) levels relative to tubulin loading control in RPE-1 cells treated with vehicle control or 200 U/mL IFNγ for 24 h. After cell lysis, indicated samples were incubated with 1 M hydroxylamine pH 7.0 for 1 h. For quantification, the 75 kDa saturated band was excluded from analysis and the signal intensity relative to tubulin loading control was normalised to IFNγ-treated cells. Mean ± SEM ($n = 3$, from three separate experiments). *$p < 0.05$.

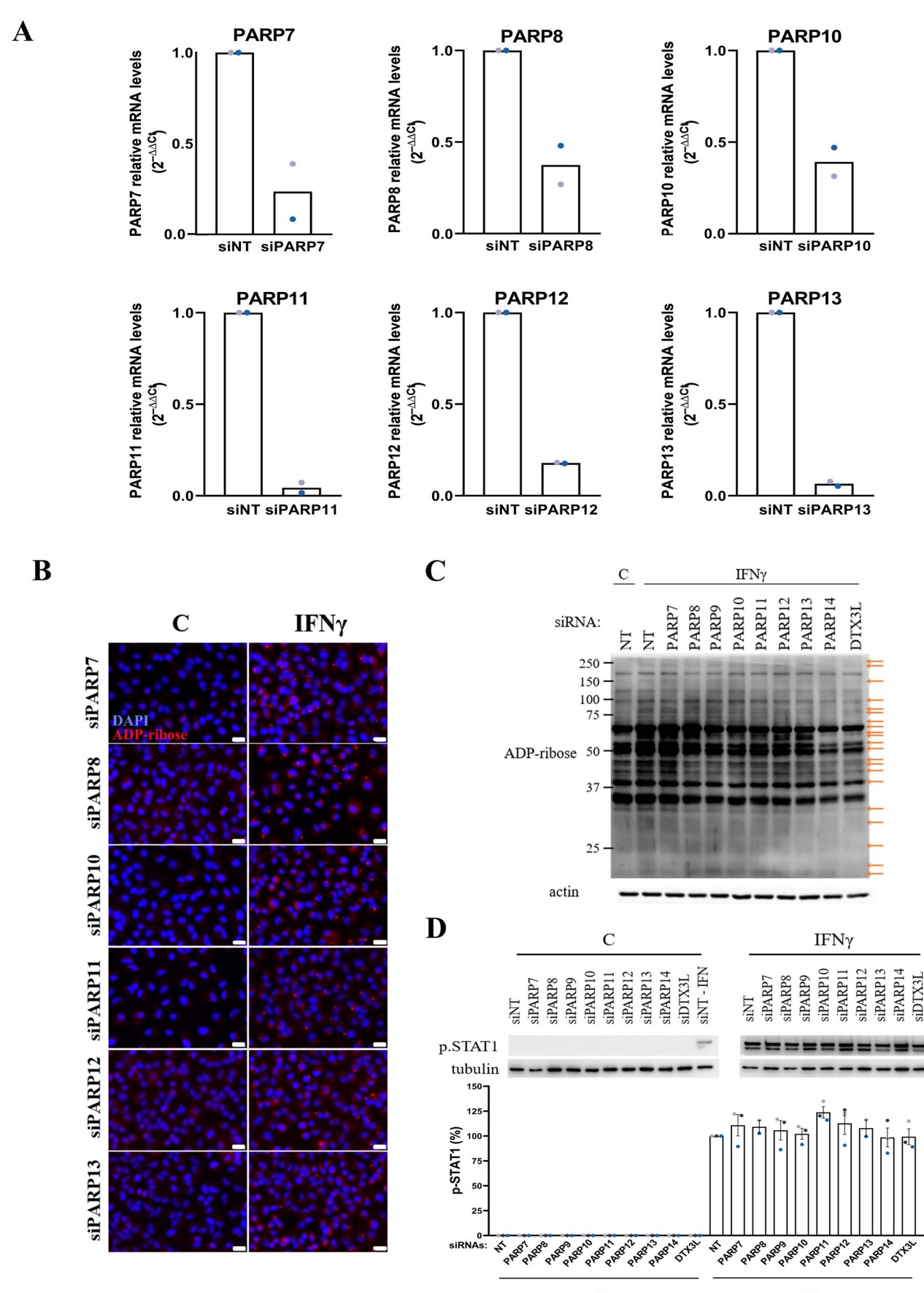

◀

**Figure EV2.   IFNγ-induced ADP-ribosylation is dependent on the PARP9/DTX3L complex and PARP14.**

(related to figure 2). (**A**) Quantification of relative PARP7, PARP8, PARP10, PARP11, PARP12 and PARP13 mRNA levels by RT-qPCR in A549 cells transfected with the indicated siRNAs, normalised to siNT transfected cells. Mean ± SEM ($n = 2$, from two separate experiments). siRNA efficiencies for PARP9, DTX3L and PARP14 are shown in Fig. 4A (**B**) Representative immunofluorescence microscopy images of pan-ADP-ribose (MABE1016) signal in A549 cells transfected with the indicated siRNAs, treated with vehicle control or 100 U/mL IFNγ for 24 h. Scale bar: 20 μm. siNT control is shown in Fig. 2A (**C**) Representative immunoblot for mono-ADP-ribose (43647 HRP-coupled) and actin loading control in A549 cells transfected with the indicated siRNAs, treated with vehicle control or 200 U/mL IFNγ for 24 h.
(**D**) Representative images (upper) and quantification (lower) of immunoblot analyses for STAT1 phospho-Y701 (p.STAT1) levels relative to tubulin loading control in A549 cells transfected with the indicated siRNAs, treated with vehicle control or 100 U/mL IFNγ for 24 h, normalised to IFNγ-treated siNT cells. Mean ± SEM ($n = 3$, from three separate experiments).

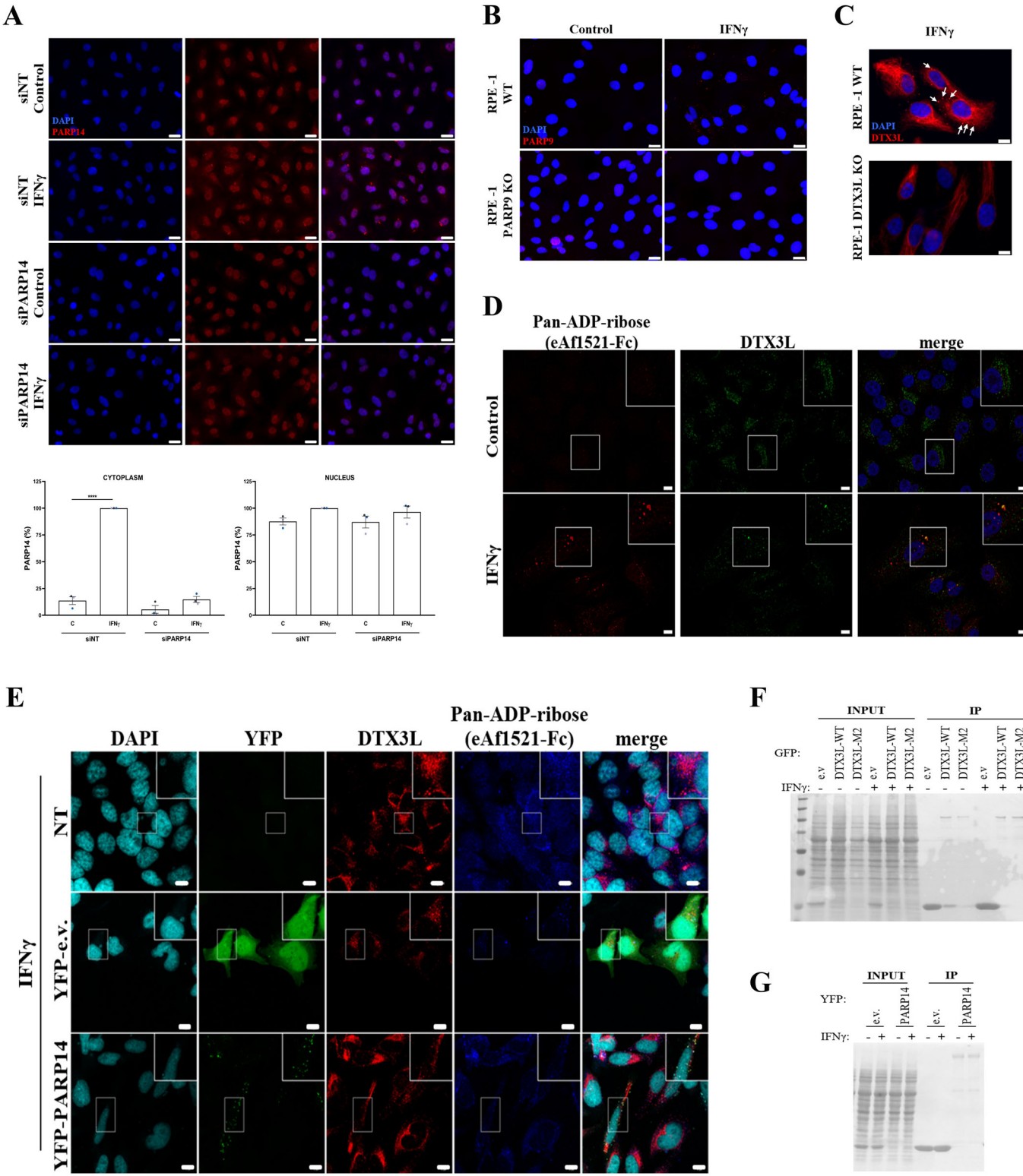

◀ **Figure EV3.** Antibody validation, IP controls and co-localization of DTX3L with PARP14 and ADP-ribose.

(related to figure 3). (**A–C**) PARP14, PARP9 and DTX3L antibody validation. (**A**) Representative immunofluorescence microscopy images (upper) and quantification (lower) of PARP14 signal in the cytoplasm (lower left) and nuclei (lower right) in A549 cells transfected with indicated siRNAs, treated with vehicle control or 100 U/mL IFNγ for 24 h. (**B**) Representative immunofluorescence microscopy images of PARP9 staining in RPE-1 WT or PARP9 knockout RPE-1 cells treated with vehicle control or 100 U/mL IFNγ for 24 h. (**C**) Representative immunofluorescence microscopy images of DTX3L staining in RPE-1 WT or DTX3L knockout RPE-1 cells treated with 100 U/mL IFNγ for 24 h. White arrows indicate the specific DTX3L cytoplasmic dots. Scale bar: 10 μm. (**D**) Representative immunofluorescence microscopy images of A549 cells treated or not with 500 U/mL IFNγ for 24 h, co-stained for pan-ADP-ribose (eAF1521-Fc) and DTX3L. Regions marked with a white box are enlarged in the top right corner. Scale bar: 10 μm. (**E**) Representative immunofluorescence confocal microscopy images of HeLa cells not transfect (NT) or transfected with YFP-empty vector (YFP-e.v.) or YFP-PARP14, treated with 200 U/mL IFNγ for 24 h, co-stained for pan-ADP-ribose (eAF1521-Fc) and DTX3L. Regions marked with a white box are enlarged in the top right corner. Scale bar: 10 μm. (**F, G**) Ponceau S staining of membranes to confirm equal loading and transfer of proteins used in Fig. 3D (**F**) and 3E (**G**).

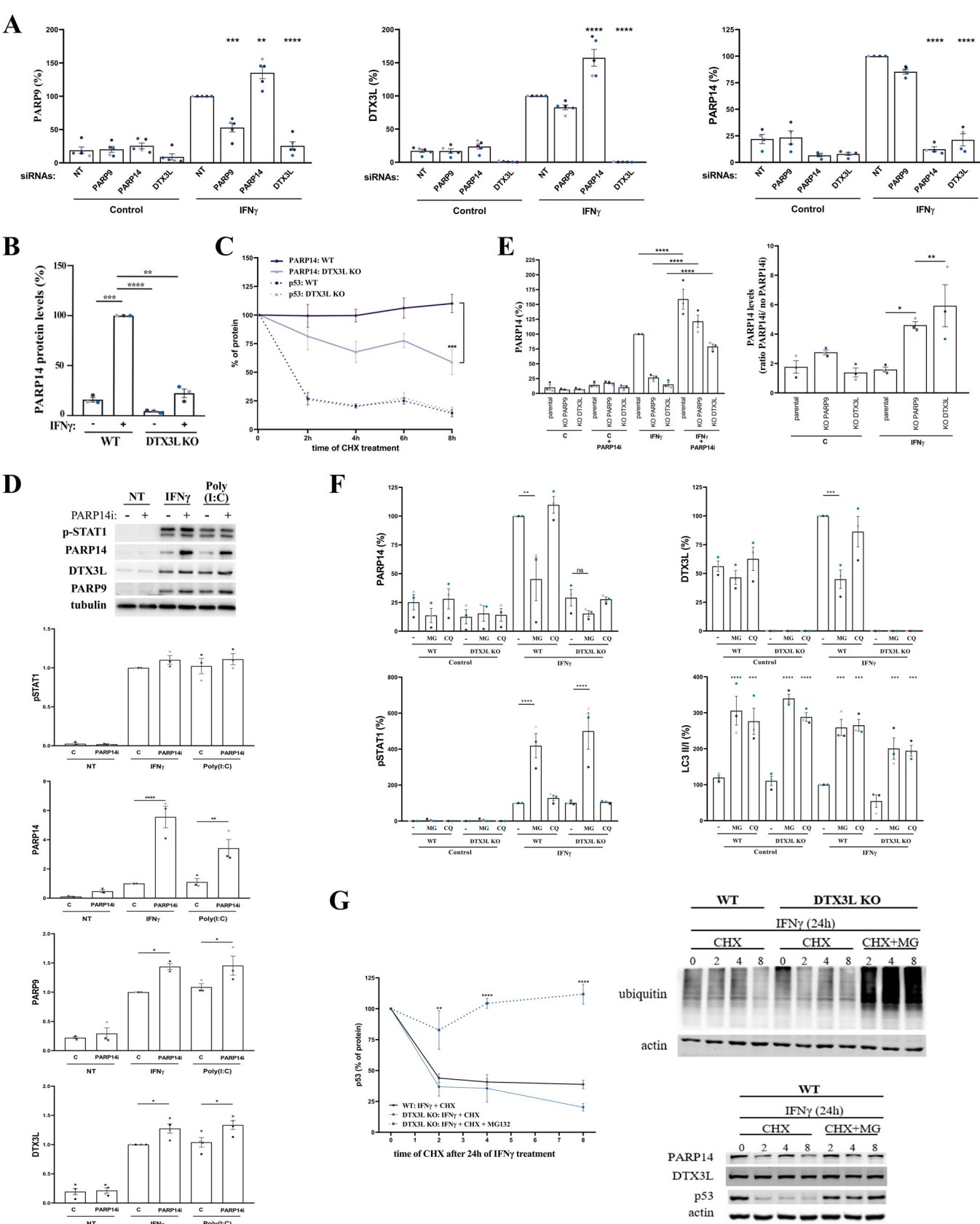

◀ **Figure EV4.   The PARP9/DTX3L complex regulates PARP14 protein stability.**

(related to figure 4). (**A**) Quantification of immunoblot analyses (as shown in Fig. 4A) for PARP9, DTX3L and PARP14 protein levels relative to tubulin loading control in A549 cells transfected with indicated siRNAs and treated with vehicle control or 100 U/ml IFNγ for 24 h, normalised to IFNγ-treated siNT cells. (**B**) Quantification of immunoblot analyses (as shown in Fig. 4B) for PARP14 protein levels relative to tubulin loading control in RPE-1 WT or DTX3L KO cells treated with vehicle control or 100 U/mL IFNγ for 24 h, normalised to IFNγ-treated WT cells. (**C**) Quantification of immunoblot analyses (as shown in Fig. 4D) for PARP14 and p53 in RPE-1 WT or DTX3L KO cells treated with 50 μg/mL cycloheximide (CHX) for the indicated times, normalised to untreated controls. (**D**) Representative image and quantification of immunoblot analyses for pSTAT1, PARP14, PARP9 and DTX3L levels relative to tubulin loading control in A549 cells, 24 h after treatment with vehicle control, 100 U/mL IFNγ, transfection with 0.1 μg/mL poly(I:C) and/or 100 nM PARP14i, normalised to IFNγ-treated cells. (**E**) Quantification of immunoblot analyses (as shown in Fig. 4E) for PARP14 protein in RPE-1 WT, PARP9 KO or DTX3L KO cells treated with vehicle control or 100 U/mL IFNγ and/or 100 nM PARP14i for 24 h, as indicated. PARP14 levels relative to tubulin loading control, normalised to IFNγ-treated WT cells (left) and the ratio between the PARP14i treated and respective non-treated samples (right) are shown. (**F**) Quantification of immunoblot analyses (shown in Fig. 4F) for PARP14, DTX3L, STAT1 phospho-Y701 (p-STAT1) levels and LC3II/LC3I ratio relative to tubulin loading control in RPE-1 WT or DTX3L KO cells treated with vehicle controls or 100 U/mL IFNγ, and 20 μM chloroquine (CQ) or 10 μM MG132 as indicated, for 24 h, normalised to IFNγ-treated WT cells. (**G**) Quantification of immunoblot analyses for p53 in RPE-1 WT or DTX3L KO (left) and representative image of immunoblot analyses for ubiquitin (upper right) and PARP14, DTX3L, p53 and actin loading control (lower right) in RPE-1 WT cells treated with 50 μg/mL cycloheximide (CHX) and 10 μM MG132 for the indicated times after treatment with 100 U/mL IFNγ for 24 h, normalised to untreated samples of each cell line. Mean ± SEM ($n = 3$–5, as indicated). *$p < 0.05$, **$p < 0.01$, ***$p < 0.001$ and ****$p < 0.0001$.

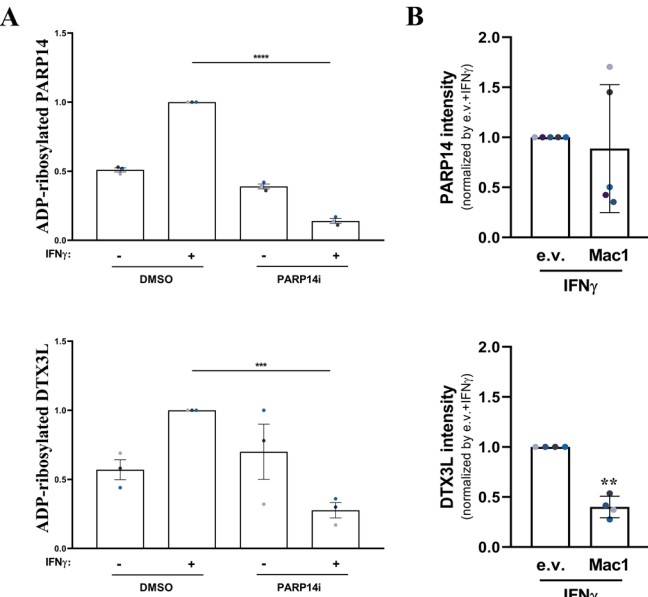

**Figure EV5. ADP-ribosylation of PARP14 and DTX3L in response to IFNγ is sensitive to Mac1 expression.**

(related to figure 5). (A) Quantification of immunoblot analyses for fluorescently co-stained mono-ADP-ribose (43647 mouse Fc-conjugated) and either PARP14 (upper) or DTX3L (lower) in A549 cells treated with vehicle control or 100 U/mL IFNγ for 24 h and/or 100 nM PARP14 inhibitor, as indicated. Graphs show the ratio between the mono-ADP-ribose band at the respective molecular weight and the total protein levels, normalised to the IFNγ-treated sample. (B) Quantification of PARP14 (upper) and DTX3L (left) immunoblot bands of GST-*Af1521* pulldown in A549 cells transduced with an empty vector (e.v.) or FLAG-tagged SARS-CoV-2 Nsp3 macrodomain (Mac1) lysates, 24 h after treatment with 100 U/mL IFNγ. Mean ± SEM ($n = 3$–5, as indicated). ***$p < 0.001$ and ****$p < 0.0001$.

