## [Peer Review File · The EMBO Journal]

PARP14 is regulated by the PARP9/DTX3L complex and promotes interferon γ -induced ADP-ribosylation

Victoria Chaves Ribeiro, Lilian Cristina Russo, and Nicolas Carlos Hoch

Corresponding author(s): Nicolas Hoch (nicolas@iq.usp.br)

Review Timeline:

Submission Date:	7th Oct 23
Pre-decision consultation:	24th Nov 23
Authors' Revision Plan:	1st Dec 23
Editorial Decision:	15th Dec 23
Revision Received:	16th Mar 24
Editorial Decision:	30th Apr 24
Revision Received:	3rd May 24
Accepted:	8th May 24

Editor: Hartmut Vodermaier

Transaction Report:

Dear Dr. Hoch,

Thank you for submitting your manuscript on PARP14 in interferon-induced ADP-ribosylation for our consideration. We have now received the reports of three expert referees, which I copy below for your information. As you will see, the referees acknowledge the potential interest and timeliness of the work, but also raise a considerable number of concerns. These issues include insufficient reagent characterization, controls and validation via complementary approaches, as well as inconclusive experiments or missing deeper insights on certain aspects. It is not clear if these issues could all be adequately addressed in a straightforward manner during a regular, single-round revision, but I would nevertheless give you a chance to consider the reports, and to get back to me with a tentative revision plan explaining what you might (and might not) be able to add, should you be given the opportunity to revise this work for The EMBO Journal. Based on such a proposal and preliminary point-by-point response to the referees' comments, I could then determine whether a major revision for The EMBO Journal would seem realistic, or whether a less substantively revised (but still mechanistically strengthened) version might at least be suitable for one of our sister journals. I'd also be happy to talk through such a revision proposal with you if needed.

Looking forward to hearing from you,

Best regards,

Hartmut Vodermaier

Referee #1 (Report for Author)

Review PARP14 papers

The manuscript by Ribeiro et al. demonstrates that PARP14 is the principal PARP family member responsible for IFN γ -stimulated cellular ADP-ribosylation. They also provide evidence that PARP9/DTX3l regulates PARP14 catalytic activity, perhaps by directly inhibiting its catalytic activity. Overall, this manuscript is well-written and timely given the growing interest in PARPs, and in particular PARP14, because the first PARP14 inhibitor is in clinical trials. However, several items need to be addressed prior to publication:

1. The authors should provide validation of their siRNA knockdowns used in this

study.

2. Fig 1B: the 33205 antibody was described as a Ser-specific ADPr detection reagent, but the authors find that the signal they get with the 33205 antibody is similar to the 33204 antibody (which detects mono-ADPr on several amino acids) (Bonfiglio et al., Cell 2020). The authors should explain this result in the text.

3. Fig. 1C: the authors state that H₂O₂-mediated nuclear ADP-ribosylation targets Ser which is why the signal doesn't diminish upon hydroxylamine treatment. However, several studies in the literature show that H₂O₂ can indeed generate Glu/Asp-ADPr modifications (e.g., Zhang et al., Nature Methods, 2013). The authors should determine if ARH3, a known Ser-ADPr hydrolase, removes this signal. The authors should also use a secondary method to confirm Glu/Asp-ADPr modification in their IF images, for example, showing that Nsp3 macro can remove the signal.

4. Fig 3: The authors should confirm the PARP14 nuclear signal using PARP14 siRNAs.

5. Fig. 4/Sup. Fig 3: The authors should determine if the PARP14 protein stability comparison between WT and DTX3L is statistically significant.

6. Fig 4f: Why are P14 levels lower with MG132/CQ?

7. Does knockdown of DTX3L alter PARP14 cytoplasmic puncta formation or nuclear protein levels? This can be assessed by IF or biochemical fractionation.

8. Given the MG132/CQ studies, is it possible that the loss of DTX3L drives PARP14 into "insoluble" aggregates due to excessive auto-mono-ADP-ribosylation of PARP14?

9. Fig. 5: the Af1521 pulldown doesn't show direct mono-ADP-ribosylation. Can the authors show this by IP of DTX3L and probing from mono-ADPr in the presence and absence of a PARP14 inhibitor?

Other minor comments:

1. Page 1, line 27: reference for PARP7 is missing.

2. Sup. Fig. 2: labeled incorrectly-should be PARP14.

3. The authors should explain why they switched to RPE-1 cells for certain experiments.

Referee #2 (Report for Author)

Riberio et al. show that addition of IFN-gamma results in increased cellular ADP-ribosylation (ADPr) as detected a microscopy-based assay. They show that this increased ADPr signal is dependent on the catalytic activity of PARP14 and can be reduced by overexpression of a macrodomain (Mac1) from SARS-CoV-2. They further show that PARP14 activity is regulated by the PARP9/DTX3L complex, although many of the mechanisms for this remain to be determined.

There are several interesting aspects to this paper. The authors provide clear data on the requirement for PARP14 in the IFN-induced increase in ADPr signal in their microscopy assay. In addition, they clearly show that PARP14 activity is regulated by PARP9/DTX3L. Both of these pieces of data further support the model that has been emerging from their work and others that PARP14, PARP9, and DTX3L are important regulators of the IFN-gamma response. However, in several places, the authors overinterpret their data, drawing conclusions that are not supported by the data, which weakens enthusiasm for the paper. Detailed comments are below:

Major concerns:

- 1) Many of the results are dependent on a single microscopy assay. Based on the western blot data shown in Figure 5B (especially the slice shown in the right panel), there is little to no increase in ADPr signal upon IFN-gamma induction, which calls into question the microscopy data. This same inconsistency across assays and/or antibodies is present in the co-submitted manuscript from Kar et al. Can the authors compare antibodies in both western blot and fluorescence assays to be fully transparent about when there is an IFN-gamma-induced increase in ADPr? Detection of this signal in an antibody-specific way does not discount its existence, but it would be useful for future studies for the authors to be clear about when this can be observed and when it can not and even provide some speculation about the reasons for those differences.
- 2) Again for full transparency, it would be appropriate for the authors to show (potentially as supplementary materials) full western blots rather than the single slices as are shown in Figure 3D or 5B. Analysis of the entire blot, rather than the slice that the authors want to focus on, will allow the reader to interpret the data.
- 3) The authors do not formally show that the ADPr signal that they detect is directly catalyzed by PARP14. Given all of the cross-regulation of PARPs, and the dependence of these enzymes on each other (some of which is nicely shown in this paper), it would be more appropriate to be interpret these results as indicating that PARP14 activity is required for this increase in ADPr signal rather than the premature assertion that it is "catalysed by PARP14".
- 4) The assertion that PARP14 and DTX3L ADP-ribosylation are the "first cellular targets of the viral macrodomain" is unsupported by their data and the literature. Many manuscripts have reported that overexpression of the SARS-CoV-2 macrodomain (Mac1) is able to reverse ADP-ribosylation on a variety of cellular targets. It is unclear how these results are different from those other overexpression data, thus these claims of novelty/importance throughout the paper should be removed.
- 5) A recent paper (<https://pubmed.ncbi.nlm.nih.gov/37695054/>) showed that a coronavirus Mac1 mutant virus is inhibited in vivo in a PARP12-dependent manner. This paper should be cited, rather than only citing work that implicates viral macrodomain activity as being important in the context of PARP14.

6) The antibody used for PARP14 in Figure 3 does not appear to be entirely specific for PARP14 given how little change there is in the overall signal relative to the western blot data (e.g. Figure 3C). The authors need to either their microscopy signal with a siRNA or KO of PARP14. signal remains in the PARP14 KO cells. Do the authors have another way to support the claim that PARP14 co-localizes with ADPr signal in those cells?

Other concerns:

1) In Figure 1, do the authors know that hydroxylamine treatment does not impact other aspects of IFN-gamma signaling? Can they show that STAT1 phosphorylation or PARP14 upregulation is intact in hydroxylamine-treated cells?

2) In the text associated with Fig 4A and Supplementary Figure 3A, the authors claim that PARP9 silencing led to a 'striking reduction in PARP14 levels'. Neither of these figures support that claim. Visually, Figure 4A shows no difference. Quantitatively, Supplementary Figure 3A does not show a significant reduction. Demonstratively false statements such as these reduce the confidence in conclusions drawn in the paper.

3) IFN-gamma is not the only type of IFN. The authors should be more specific when referring to "IFN" when they only test IFN-gamma.

Referee #3 (Report for Author)

This is an interesting manuscript that seeks to understand the role of DTX3L/PARP9 in modulating PARP14 levels/biochemical output. The biological interest in this question stems in part from the observation that PARP14 appears to be responsible for the bulk of ADP-ribosylation that occurs in response to IFN treatment, which seems logical given that PARP14 itself is induced robustly by IFN. Unfortunately, the authors never answer the question of how DTX3L contributes to PARP14 levels/activity in the cell. The argument for a post-transcriptional mechanism is not well-founded using the data provided. The finding that PARP14 binding to AF1521 beads can be reduced by over-expression of the CoV-2 Nsp3 macrodomain would be intriguing if Nsp3 modulates PARP14 ADP-ribosylation of cellular targets relevant to a viral response. As presented, the Nsp data might also be accounted for by competing with AF1521 for binding ADP-ribosylated substrates including auto-modified PARP14.

Specific comments

Figure 1: The fluorescence imaging is intended to show that IFN signaling induces mono-ADP-ribosylation on acidic residues in target proteins. The probe used to draw this conclusion, eAF1521, is a pan-ADP-ribose detection reagent that binds both mono- and poly-ADP-ribosylated substrates, so it is hard to

know what fraction of each is induced by IFN. The authors have access to mono-specific reagents (Fig. 1A, B), but as far as I can tell the mono-specific reagents are not used in the paper beyond the first figure. Presumably IFN-induced ADP-ribosylation on hydroxylamine-sensitive sites is a novel observation; the impact of this observation would be enhanced by quantification of the fluorescent images, and by parallel biochemical analysis (blotting; quantification).

Figure 2: The authors used a focused siRNA screen and found that PARP14 mediates IFN-induced ADP-ribosylation in cells - which also is sensitive to the PARP14i from Ribon Inc. The data would be strengthened with a parallel biochemical analysis since the spots detected in the microscope are of unknown etiology, spare the fact they are induced by IFN.

Figure 3: The authors look for interactions between PARP14 and DTX3L/PARP9 after inferring both localize to small cytoplasmic foci labeled by eAF1521. Two points here - the authors should consider imaging epitope-tagged PARP14 and PARP9 in the same cells if the basic question is one of co-localization, since this will enable asking the question without the complication of changes in PARP14 expression induced by IFN. Second, the PARP14 immunoprecipitation is so inefficient I would be very cautious about interpreting this negative result. Overall, not much is learned from this figure since it is based on a negative result.

Figure 4: The key panels in this figure (4C, D) attempt to address how DTX3L affects PARP14 expression. The data as presented do not permit a clear assessment. The authors suggest a transcriptional effect of DTX3L was ruled out, but the "non-significant" difference in RPE-1 WT vs. RPE-1 DTX3L KO might be explained by replicate variance. The CHX experiment addressing PARP14 half-life is not readily interpretable without quantification. Ideally the PARP14 half-life measurements will be made after IFN treatment given that PARP14 activity seems most relevant in the context of IFN signaling. The MG132 and CLQ data as presented (neg results) without positive controls cannot be used to rule out proteasome and autophagy involvement; it would be appropriate to test if/how the protein half-life is affected by these treatments since the result seems surprising. The biochemical fractionation (4G) is well done but I can't judge if this contributes to the understanding.

Figure 5: The authors show that Nsp overexpression reduces the amount of PARP14 and DTX3L captured on AF1521 beads. Since the question is whether the viral macrodomain hydrolyzes ADP-ribose from these proteins, it would seem prudent for the authors to IP PARP14 and DTX3L separately and probe for ADP-ribose on blots as a more direct test. Otherwise, there are multiple possibilities for how Nsp expression might affect protein recovery on AF1521 beads using cell lysates.

Referee #1 (Report for Author)

Review PARP14 papers

The manuscript by Ribeiro et al. demonstrates that PARP14 is the principal PARP family member responsible for IFN γ -stimulated cellular ADP-ribosylation. They also provide evidence that PARP9/DTX3L regulates PARP14 catalytic activity, perhaps by directly inhibiting its catalytic activity. Overall, this manuscript is well-written and timely given the growing interest in PARPs, and in particular PARP14, because the first PARP14 inhibitor is in clinical trials. However, several items need to be addressed prior to publication:

1. The authors should provide validation of their siRNA knockdowns used in this study.

We will provide a short explanation containing the following info:

- this is a screen and the procedure we use works well for most tested siRNAs
- knockdown efficiency of siPARP9, siDTX3L and siPARP14 is validated by western blotting elsewhere
- the effect of these three hits on ADP-ribosylation was validated by KO or inhibitor treatment elsewhere, showing that it was a highly effective screen
- we don't have antibodies for most of these PARPs (we are unaware of good antibodies for some of them in the field as a whole)

Action: Nevertheless, we will perform RT-qPCR to demonstrate knockdown efficiency of the genes not subsequently tested by western (Exp.1)

2. Fig 1B: the 33205 antibody was described as a Ser-specific ADPr detection reagent, but the authors find that the signal they get with the 33205 antibody is similar to the 33204 antibody (which detects mono-ADPr on several amino acids) (Bonfiglio et al., Cell 2020). The authors should explain this result in the text.

We will provide a detailed explanation describing how both the data and text in the Bonfiglio et al, Cell 2020 paper actually do not show that 33205 is serine-ADPr-specific. Instead, they found that 33204 has a surprising preference for "non-serine" modification over serines, whereas 33205 recognizes both serines and "non-serine" modification equally well. Therefore 33205 is better suited to detect Ser linkage, but can also detect other modifications, which may have led to confusion. The fact that 33205 is not Ser-specific and can detect PARP14-catalysed modification is nicely illustrated in a supplementary figure in the same Bonfiglio paper showing that both 33204 and 33205 can detect auto-modified PARP14. Therefore, one cannot draw any conclusions in terms of the ADPr linkage using these reagents alone.

Action: Include a sentence on this in the discussion.

3. Fig. 1C: the authors state that H₂O₂-mediated nuclear ADP-ribosylation targets Ser which is why the signal doesn't diminish upon hydroxylamine treatment. However, several studies in the literature show that H₂O₂ can indeed generate Glu/Asp-ADPr modifications (e.g., Zhang et al., Nature Methods, 2013). The authors should determine if ARH3, a known Ser-ADPr hydrolase, removes this signal. The authors should also

use a secondary method to confirm Glu/Asp-ADPr modification in their IF images, for example, showing that Nsp3 macro can remove the signal.

We will provide a detailed explanation of the following points:

- The peroxide-treated sample is only a positive control to show that hydroxylamine does not prevent detection of ADPr by immunofluorescence in a non-specific manner, as we are unaware of any lab that has performed this technique on coverslips before
- The argument in the field on whether PARP1 modifies Glu/Asp or Serines (and the relative proportion of them) is not relevant to this manuscript, although this novel technique may be useful to answer this question elsewhere
- The peroxide-induced signal has been repeatedly shown in the literature to persist in ARH3 KO cells, showing that at least a large portion of it is Ser-targeted (include citations).
- We have previously shown in (Russo et al JBC 2021) that the IFN-induced signal is sensitive to ectopic expression of the Nsp3 macrodomain in cells, including the use of a catalytically-dead version of the domain, which did not hydrolyse the modification, which in itself already suggested it should be targeted to Glu/Asp residues.
- The request for independent confirmation that the IFN-induced signal is on Glu/Asp is valid, and we will perform the more traditional hydroxylamine treatment on soluble proteins followed by western blotting using the mono-ADPr reagent used in Fig. 5B, as also requested by reviewer 3, major comment 1.

Actions:

- Amend the text to soften the sentence stating that PARP1 modifies serine, to allow the possibility that some of the signal is also on Glu/Asp.
- Perform hydroxylamine treatment on cell lysates, followed by anti-ADPr western to confirm hydroxylamine-sensitivity of the IFN-induced ADPr signal (Exp.2)

4. Fig 3: The authors should confirm the PARP14 nuclear signal using PARP14 siRNAs.

We will provide a short explanation that this has already been done in Sup. Fig. 2A, which may have been missed because the figure was unfortunately mislabelled (as spotted by this reviewer in their minor comment 2 below).

Action: Amend Sup. Fig 2A labelling

25. Fig. 4/Sup. Fig 3: The authors should determine if the PARP14 protein stability comparison between WT and DTX3L is statistically significant.

This has already been done and is statistically significant.

Action: Include statistical analysis of Sup.Fig. 3C

6. Fig 4f: Why are P14 levels lower with MG132/CQ?

Short explanation that this was also surprising to us. Could be caused by an effect of MG132 and CQ on interferon signalling itself. This possibility will be checked by determining IFN-induced STAT1 phosphorylation in the presence of MG132 or CQ.

Action: Completely repeat this experiment, including additional blots for phospho-STAT1 (to test if MG132 or CQ impair IFN signalling) and LC3 and ubiquitin (positive controls that CQ and MG132 worked, respectively – requested by reviewer 3 comment 4) (Exp. 3)

7. Does knockdown of DTX3L alter PARP14 cytoplasmic puncta formation or nuclear protein levels? This can be assessed by IF or biochemical fractionation.

The suggested biochemical fractionation has been performed in DTX3L KO cells and is shown in Fig. 4G (right panel). We will perform the IF experiment as well.

Action: Perform anti-PARP14 immunofluorescence in WT and DTX3L KO cells --+ IFN (Exp. 4)

8. Given the MG132/CQ studies, is it possible that the loss of DTX3L drives PARP14 into "insoluble" aggregates due to excessive auto-mono-ADP-ribosylation of PARP14?

Short explanation that our sample preparation, which involves lysing cells directly in pre-heated Laemmli buffer containing SDS, precludes this possibility.

9. Fig. 5: the Af1521 pulldown doesn't show direct mono-ADP-ribosylation. Can the authors show this by IP of DTX3L and probing from mono-ADPr in the presence and absence of a PARP14 inhibitor?

Short explanation that the fluorescent ADPr western blot (Fig. 5B) is already an independent confirmation of the Af1521 pulldown. However, we will perform the IP-WB as also requested by reviewer #3, comment 5.

Action: Perform IP of DTX3L and PARP14 in cells treated or not with IFN γ , in presence or absence of PARP14i and expressing or not the Nsp3 macrodomain, blotting for mono-ADPr using AbD43647 (Exp. 5)

Other minor comments:

1. Page 1, line 27: reference for PARP7 is missing.

Action: Correct the sentence/citation

2. Sup. Fig. 2: labeled incorrectly-should be PARP14.

Action: correct the labelling

3. The authors should explain why they switched to RPE-1 cells for certain experiments.

Explanation describing following points:

- We previously showed that IFN γ -induced ADPr can be observed in both A549 and RPE1 cells, so the response is not limited to particular cell lines, allowing confirmation of results between them.
- These cell lines are from different tissues and A549 is from a tumor, while RPE1 is a normal diploid cell line immortalized by hTERT, removing caveats from the oncogenic process.
- We have A549 cells expressing Nsp3 macrodomain, and PARP9 and DTX3L KO in the RPE background, generated in the previous publication (Russo et al 2021)
- For co-localization studies, we preferred the A549 cells, as the background cytosolic ADPr signal in RPE1 cells is a bit higher with the eAF1521 reagent, which is the only reagent detected by secondary mouse antibody, and is therefore necessary to co-stain ADPr with all other antibodies raised in rabbit.

Action: Add a sentence on the use of these cell lines in the discussion.

Referee #2 (Report for Author)

Riberio et al. show that addition of IFN-gamma results in increased cellular ADP-ribosylation (ADPr) as detected a microscopy-based assay. They show that this increased ADPr signal is dependent on the catalytic activity of PARP14 and can be reduced by overexpression of a macrodomain (Mac1) from SARS-CoV-2. They further show that PARP14 activity is regulated by the PARP9/DTX3L complex, although many of the mechanisms for this remain to be determined.

There are several interesting aspects to this paper. The authors provide clear data on the requirement for PARP14 in the IFN-induced increase in ADPr signal in their microscopy assay. In addition, they clearly show that PARP14 activity is regulated by PARP9/DTX3L. Both of these pieces of data further support the model that has been emerging from their work and others that PARP14, PARP9, and DTX3L are important regulators of the IFN-gamma response. However, in several places, the authors overinterpret their data, drawing conclusions that are not supported by the data, which weakens enthusiasm for the paper. Detailed comments are below:

Major concerns:

1) Many of the results are dependent on a single microscopy assay. Based on the western blot data shown in Figure 5B (especially the slice shown in the right panel), there is little to no increase in ADPr signal upon IFN-gamma induction, which calls into question the microscopy data. This same inconsistency across assays and/or antibodies is present in the co-submitted manuscript from Kar et al. Can the authors compare antibodies in both western blot and fluorescence assays to be fully transparent about when there is an IFN-gamma-induced increase in ADPr? Detection of this signal in an antibody-specific way does not discount its existence, but it would be useful for future studies for the authors to be clear about when this

can be observed and when it can not and even provide some speculation about the reasons for those differences.

Detailed explanation on the following topics:

- The IFN-induced ADPr signal is detectable with four different anti-ADPr detection reagents (2x pan-ADPr, 2x mono-ADPr) by immunofluorescence, by western blotting using a new (and more sensitive) mono-ADPr reagent (AbD43647) and by Af1521 pulldown.
- Several groups (including Ahel in accompanying paper) have replicated this observation
- The AbD43647 reagent (also used in the accompanying Ahel paper) is the only one that can detect IFN-induced ADPr by western blot in our hands.
- IFN-induced ADPr is weaker than the DNA damage-induced PARP1 signal, so the IFN signal is much more reliant on sample preparation and quality of the ADPr reagents
- There is indeed a need for better understanding of the specificities of these reagents and their applicability for different methodologies (not just for this study). We are preparing a manuscript on this subject.
- We can speculate on two reasons why the IF signal is easier to detect than by western blot. One is that IFN-induced ADPr may also target RNA (cite recent Ahel papers), the other is that Glu/Asp-targeted ADPr is known to be labile (cite a few references on this), and may be partially lost during sample preparation for western blots.

2) Again for full transparency, it would be appropriate for the authors to show (potentially as supplementary materials) full western blots rather than the single slices as are shown in Figure 3D or 5B. Analysis of the entire blot, rather than the slice that the authors want to focus on, will allow the reader to interpret the data.

I believe raw western blot images are a standard requirement for EMBO, so we will provide them for the revision, including some full molecular weight range mono-ADPr blots we already have. We do however routinely cut the membranes to approximate expected sizes of target proteins, to allow simultaneous detection of multiple proteins on the same sample/membrane, as stripping/re-probing can lead to sample loss. During experiments for the revision process, we will avoid cutting membranes to allow the presentation of full molecular weight range ADPr blots. (as also shown in the accompanying paper by the Ahel group)

Action: Provide raw figures of all western blots. Provide full molecular range blots for mono-ADPr.

3) The authors do not formally show that the ADPr signal that they detect is directly catalyzed by PARP14. Given all of the cross-regulation of PARPs, and the dependence of these enzymes on each other (some of which is nicely shown in this paper), it would be more appropriate to be interpret these results as indicating that PARP14 activity is required for this increase in ADPr signal rather than the premature assertion that it is "catalysed by PARP14".

Short discussion pointing out that the, while the reviewer is correct that we cannot categorically say that PARP14 is the enzyme catalysing all of this modification, the most parsimonious interpretation of our data (and that of the accompanying paper)

is that PARP14 directly catalyses this ADP-ribosylation. Nevertheless, we will soften the wording.

Action: Change wording throughout the manuscript to soften statements such as “PARP14-catalysed” to “PARP14-dependent”.

4) The assertion that PARP14 and DTX3L ADP-ribosylation are the "first cellular targets of the viral macrodomain" is unsupported by their data and the literature. Many manuscripts have reported that overexpression of the SARS-CoV-2 macrodomain (Mac1) is able to reverse ADP-ribosylation on a variety of cellular targets. It is unclear how these results are different from those other overexpression data, thus these claims of novelty/importance throughout the paper should be removed.

Unfortunately the reviewer did not provide references for the manuscripts in which SARS-CoV-2 macrodomain overexpression was said to have been performed. We are aware of many publications showing that the Nsp3 macrodomain can hydrolyse various targets in *in vitro* biochemical reactions, but at the moment we stand by the statement that this is the first study to show that the SARS-CoV-2 macrodomain hydrolyses a specific target in cells.

Action: We will provide a thorough manual analysis of the literature, listing viral macrodomain targets identified both in biochemical assays and in cells to refine and substantiate our claim. According to this analysis, we will adjust the statement to “viral macrodomain”, “coronavirus macrodomain”, “SARS-CoV-2 macrodomain” or remove the claim altogether.

5) A recent paper (<https://pubmed.ncbi.nlm.nih.gov/37695054/>) showed that a coronavirus Mac1 mutant virus is inhibited *in vivo* in a PARP12-dependent manner. This paper should be cited, rather than only citing work that implicates viral macrodomain activity as being important in the context of PARP14.

Highlighting the current interest in ADP-ribosylation in antiviral responses, this paper was published only weeks before initial submission of our manuscript, but should indeed be included in the discussion.

Action: include a sentence referencing the findings of this article

6) The antibody used for PARP14 in Figure 3 does not appear to be entirely specific for PARP14 given how little change there is in the overall signal relative to the western blot data (e.g. Figure 3C). The authors need to either their microscopy signal with a siRNA or KO of PARP14. signal remains in the PARP14 KO cells. Do the authors have another way to support the claim that PARP14 co-localizes with ADPr signal in those cells?

Similar to (reviewer #1, comment 4) this has already been done and is shown in Sup. Fig. 2A, which was unfortunately mislabelled.

To support the co-localization between PARP14 and ADPr, and between PARP14 and PARP9/DTX3L (raised by reviewer #3, comment 3) we will determine localization of overexpressed YFP-PARP14 already available in the lab relative to endogenous ADPr and PARP9/DTX3L.

Action: Overexpress YFP-PARP14 and determine co-localization with ADPr, PARP9 or DTX3L by immunofluorescence microscopy (Exp. 6)

Other concerns:

1) In Figure 1, do the authors know that hydroxylamine treatment does not impact other aspects of IFN-gamma signaling? Can they show that STAT1 phosphorylation or PARP14 upregulation is intact in hydroxylamine-treated cells?

Short explanation that the hydroxylamine incubation is a chemical treatment of the samples after fixation and permeabilization of the samples, so cells are dead at that point.

2) In the text associated with Fig 4A and Supplementary Figure 3A, the authors claim that PARP9 silencing led to a 'striking reduction in PARP14 levels'. Neither of these figures support that claim. Visually, Figure 4A shows no difference. Quantitatively, Supplementary Figure 3A does not show a significant reduction. Demonstratively false statements such as these reduce the confidence in conclusions drawn in the paper.

This was an unfortunate grammatical mistake. The "striking reduction" was meant to refer to the siDTX3L sample, without extending to siPARP9, but the sentence as written did convey this erroneous idea. This will be amended. However, in other data (Fig. 4E and 4G) we show that PARP9 KO does indeed reduce PARP14 levels quite substantially, to similar levels as DTX3L KO. Therefore, the milder effect of siPARP9 on PARP14 levels may reflect a reduced efficiency of PARP9 depletion in this experiment compared to the DTX3L knockdown.

Action: Amend the sentence as suggested

3) IFN-gamma is not the only type of IFN. The authors should be more specific when referring to "IFN" when they only test IFN-gamma.

Short mention that we have shown previously (Russo et al 2021) that ADP-ribosylation can also be induced by IFN-alpha, IFN-beta and poly(I:C), but signal is always strongest with IFN-gamma, which is why we use it for most experiments.

Action: Revise the manuscript to replace "IFN" with "IFN-gamma" where appropriate.

Referee #3 (Report for Author)

This is an interesting manuscript that seeks to understand the role of DTX3L/PARP9 in modulating PARP14 levels/biochemical output. The biological interest in this question stems in part from the observation that PARP14 appears to be responsible for the bulk of ADP-ribosylation that occurs in response to IFN treatment, which seems logical given that PARP14 itself is induced robustly by IFN. Unfortunately, the authors never answer the question of how DTX3L contributes to PARP14 levels/activity in the cell. The argument for a post-transcriptional mechanism is not well-founded using the data provided. The finding that PARP14 binding to AF1521 beads can be reduced by over-expression of the CoV-2 Nsp3 macrodomain would be intriguing if Nsp3 modulates PARP14 ADP-ribosylation of

cellular targets relevant to a viral response. As presented, the Nsp data might also be accounted for by competing with AF1521 for binding ADP-ribosylated substrates including auto-modified PARP14.

Specific comments

Figure 1: The fluorescence imaging is intended to show that IFN signaling induces mono-ADP-ribosylation on acidic residues in target proteins. The probe used to draw this conclusion, eAF1521, is a pan-ADP-ribose detection reagent that binds both mono- and poly-ADP-ribosylated substrates, so it is hard to know what fraction of each is induced by IFN. The authors have access to mono-specific reagents (Fig. 1A, B), but as far as I can tell the mono-specific reagents are not used in the paper beyond the first figure. Presumably IFN-induced ADP-ribosylation on hydroxylamine-sensitive sites is a novel observation; the impact of this observation would be enhanced by quantification of the fluorescent images, and by parallel biochemical analysis (blotting; quantification).

Detailed discussion of the following points:

- The conclusion that the modification is mono-ADPr is drawn from the data using mono-ADPr-specific reagents (AbD33240 and AbD33205), not the eAF1521 pan-ADPr reagent (Fig. 1A, 1B). These figures show not only that two different mono-ADPr reagents can detect IFN-induced ADP-ribosylation, but also that the mono-ADPr signals co-localize with the pan-ADPr signal, showing that it is specific and the same structure can be detected by all of these different reagents.
- These mono-ADPr reagents have only recently become commercially available, so most experiments were performed with a pan-ADPr reagent. As shown in Fig1A and 1B, these signals co-localize extensively, so the remaining figures are reporting on the same structure/event.
- The western blot in Fig. 5B was performed with another mono-ADPr-specific reagent (AbD43647), confirming this observation with a different method and reagent
- As also suggested by reviewer #1, comment 3, we will perform hydroxylamine treatment on cell lysates, followed by western blot using the new mono-ADPr reagent (AbD33467)

Action: Perform hydroxylamine treatment on cell lysates, followed by anti-ADPr western to confirm hydroxylamine-sensitivity of the IFN-induced ADPr signal (Exp.2 described above)

Figure 2: The authors used a focused siRNA screen and found that PARP14 mediates IFN-induced ADP-ribosylation in cells - which also is sensitive to the PARP14i from Ribon Inc. The data would be strengthened with a parallel biochemical analysis since the spots detected in the microscope are of unknown etiology, spare the fact they are induced by IFN.

Similar to the point above, the AbD43647 reagent has only recently been made commercially available and was not available at the time the siRNA screen was performed. The effect of PARP14 inhibitor on the IFN-induced signal is demonstrated by western blotting using this reagent in Fig. 5B. However, we

will confirm the screen result by western blotting also, now that the reagent is available.

Action: Repeat the siRNA screen using new AbD43647 reagent for western blotting (Exp. 7)

Figure 3: The authors look for interactions between PARP14 and DTX3L/PARP9 after inferring both localize to small cytoplasmic foci labeled by eAF1521. Two points here - the authors should consider imaging epitope-tagged PARP14 and PARP9 in the same cells if the basic question is one of co-localization, since this will enable asking the question without the complication of changes in PARP14 expression induced by IFN. Second, the PARP14 immunoprecipitation is so inefficient I would be very cautious about interpreting this negative result. Overall, not much is learned from this figure since it is based on a negative result.

Short discussion of the caveats of overexpressing epitope-tagged proteins and the lack of good antibodies made from different animal sources that allow co-localization studies (most antibodies used here are made in rabbit).

The co-IP data indeed does not demonstrate that PARP14 cannot interact with PARP9/DTX3L, as a negative result is always hard to prove, but we stand by the statement that under conditions in which the interaction between PARP9/DTX3L is clearly observed, we cannot identify an interaction with PARP14. We will use overexpressed YFP-PARP14 to perform a higher efficiency IP to test this further

Action:

-YFP-PARP14 co-localization experiment as requested by reviewer #2, comment 6 (Exp. 6 described above)

-Express YFP-PARP14 in HEK293T cells and perform GFP-trap pulldown (similar to Fig. 3D) (Exp. 8)

Figure 4: The key panels in this figure (4C, D) attempt to address how DTX3L affects PARP14 expression. The data as presented do not permit a clear assessment. The authors suggest a transcriptional effect of DTX3L was ruled out, but the "non-significant" difference in RPE-1 WT vs. RPE-1 DTX3L KO might be explained by replicate variance. The CHX experiment addressing PARP14 half-life is not readily interpretable without quantification. Ideally the PARP14 half-life measurements will be made after IFN treatment given that PARP14 activity seems most relevant in the context of IFN signaling. The MG132 and CLQ data as presented (neg results) without positive controls cannot be used to rule out proteasome and autophagy involvement; it would be appropriate to test if/how the protein half-life is affected by these treatments since the result seems surprising. The biochemical fractionation (4G) is well done but I can't judge if this contributes to the understanding.

Discussion of the following points:

- the lack of a difference in the PARP14 RT-qPCR data between WT and DTX3L KO cells is in full agreement with the data in the accompanying manuscript by the Ahel group.
- the quantification of the CHX experiment is provided in Sup. Fig. 3C

- The biochemical fractionation is important to show that a portion of PARP14 is nuclear, which could not be determined by immunofluorescence because of non-specific staining discussed above. This experiment was requested by (reviewer #1, comment 7) and helps to reconcile our data with the literature that indicates a nuclear role for PARP14 (G. Moldovan papers and Caprara paper).

Actions:

- more replicates for the RT-qPCR experiment (Fig. 4C) (Exp. 9)
- Perform an experiment in which WT or DTX3L KO cells are treated for 24h with IFN (to induce PARP14 expression) before addition of CHX, CHX+CQ or CHX+MG132, to follow PARP14 half-life in IFN-treated cells, and to determine the effect of CQ and MG132 on that half-life. (Exp. 10)
- In both Exp. 10 and Exp. 3 (reviewer #1, comment 6) we will probe for LC3 and ubiquitin to determine if CQ and MG132 are inhibiting autophagy and the proteasome, respectively.

Figure 5: The authors show that Nsp overexpression reduces the amount of PARP14 and DTX3L captured on AF1521 beads. Since the question is whether the viral macrodomain hydrolyzes ADP-ribose from these proteins, it would seem prudent for the authors to IP PARP14 and DTX3L separately and probe for ADP-ribose on blots as a more direct test. Otherwise, there are multiple possibilities for how Nsp expression might affect protein recovery on AF1521 beads using cell lysates.

Similar to (reviewer #1, comment 9) this IP-WB experiment will be performed

Action: IP-WB (Exp. 5)

Dr. Nicolas Carlos Hoch
University of Sao Paulo
Biochemistry
Av Prof. Lineu Prestes 748
Sao Paulo, Sao Paulo 05508-000
Brazil

15th Dec 2023

Re: EMBOJ-2023-115815
PARP14-catalysed ADP-ribosylation regulated by PARP9/DTX3L is a target of the viral macrodomain

Dear Nicolas,

Thank you again for sending me a tentative point-by-point response and revision plan for your recent EMBOJ submission, and apologies for the delay in getting back to you on it. I have now had a chance to consider your answers, and realized that many of the referees' major concerns appear to be addressable with straightforward experiments and clarifications. One additional point that I would still consider important is to include Western Blot validation of PARP9, PARP14, and DTX3L knockdown in response to ref 1 pt 1, instead of just referring to "validation elsewhere" (where?). Other than that, I feel that the specific experiments outlined in your letter should indeed help to make the study a compelling candidate for EMBO J publication.

I am therefore inviting you to prepare and resubmit a manuscript revised and extended along these lines - happily within a revision period extended beyond the default three months (during which publication of competing manuscript would, as per our policies, have no effect on our final decision on your study). In any case, please do keep me updated about the progress of your revision work or in case of unexpected problems with some of the experiments, to discuss further proceedings.

I should remind you that our policy to allow only a single round of (major) revision makes it important to carefully revise and answer all points raised to the referees' satisfaction at this point. Finally, please note the detailed information and guidelines on how to prepare a revision below (and in our online Guide to Authors) - closely adhering to them shall greatly facilitate the editorial process at the time of resubmission.

Thank you again for the opportunity to consider this work, and I look forward to receiving your revision in due time.

With kind regards,

Hartmut

3) Revised manuscript text (including main tables, and figure legends for main and EV figures) has to be submitted as editable

text file (e.g., .docx format). We encourage highlighting of changes (e.g., via text color) for the referees' reference.

4) Each main and each Expanded View (EV) figure should be uploaded as individual production-quality files (preferably in .eps, .tif, .jpg formats). For suggestions on figure preparation/layout, please refer to our Figure Preparation Guidelines:

8) Please note that supplementary information at EMBO Press has been superseded by the 'Expanded View' for inclusion of additional figures, tables, movies or datasets; with up to five EV Figures being typeset and directly accessible in the HTML version of the article. For details and guidance, please refer to:

embopress.org/page/journal/14602075/authorguide#expandedview

9) Digital image enhancement is acceptable practice, as long as it accurately represents the original data and conforms to community standards. If a figure has been subjected to significant electronic manipulation, this must be clearly noted in the figure legend and/or the 'Materials and Methods' section. The editors reserve the right to request original versions of figures and the original images that were used to assemble the figure. Finally, we generally encourage uploading of numerical as well as gel/blot image source data; for details see: embopress.org/page/journal/14602075/authorguide#sourcedata

At EMBO Press, we ask authors to provide source data for the main manuscript figures. Our source data coordinator will contact you to discuss which figure panels we would need source data for and will also provide you with helpful tips on how to upload and organize the files.

Further information is available in our Guide For Authors:

In the interest of ensuring the conceptual advance provided by the work, we recommend submitting a revision within 3 months (14th Mar 2024). Please discuss the revision progress ahead of this time with the editor if you require more time to complete the revisions. Use the link below to submit your revision:

Link Not Available

Referee #1

The manuscript by Ribeiro et al. demonstrates that PARP14 is the principal PARP family member responsible for IFN γ -stimulated cellular ADP-ribosylation. They also provide evidence that PARP9/DTX3L regulates PARP14 catalytic activity, perhaps by directly inhibiting its catalytic activity. Overall, this manuscript is well-written and timely given the growing interest in PARPs, and in particular PARP14, because the first PARP14 inhibitor is in clinical trials. However, several items need to be addressed prior to publication:

1. The authors should provide validation of their siRNA knockdowns used in this study.

Thank you for this suggestion. The knockdown efficiency of PARP9, DTX3L and PARP14 is demonstrated by western blotting in Figure 4A and quantified in Fig. EV4A. We have included a new Fig. EV2A, containing the results of RT-qPCR analyses of the mRNAs of each of the remaining genes in the siRNA screen, showing that all of them are efficiently depleted using our conditions. We have amended the manuscript text to reflect this change (lines 105-107) and added reference to the western blot validations in the figure legends to Fig. EV2A.

2. Fig 1B: the 33205 antibody was described as a Ser-specific ADPr detection reagent, but the authors find that the signal they get with the 33205 antibody is similar to the 33204 antibody (which detects mono-ADPr on several amino acids) (Bonfiglio et al., Cell 2020). The authors should explain this result in the text.

Thank you for this pertinent comment, which allows us to clarify what we believe is a misconception in the field. Upon close inspection of the data presented in the (Bonfiglio et al., Cell, 2020) paper, we have come to the conclusion that AbD33205 is not actually strictly Ser-specific, but that both AbD33204 and AbD33205 are able to detect "non-Ser" modifications. The view that AbD33205 is "Ser-specific" (which by the way is not stated in that study) may stem from the fact that it is a preferable reagent over the AbD33204 antibody for detection of serine mono-ADPr, as AbD33204 apparently prefers "non-Ser" modifications. For example, (Bonfiglio et al., Cell, 2020) show in Figures 4C and 4D (pasted below) that AbD33205 can detect mono-ADPr on Glu/Asp catalysed by PARP1-E988Q (in the absence of HPF1), albeit less efficiently than AbD33204. In fact, our interpretation of these figures is that AbD33204 has a preference for Glu/Asp over serine (left panel of Figure 4D), and that AbD33205 actually has a more balanced preference for different amino acid modifications (middle panel of Figure 4D). The result in Figure 4C below, which at first sight seems to indicate that AbD33204 is balanced (top panel) and AbD33205 doesn't recognize Glu/Asp well (middle panel), is actually slightly misleading, as it is evident from the pan-ADPr signal (bottom panel) that mono-Ser modification by PARP1-E988Q in the presence of HPF1 (right lane) was more efficient in this experiment than PARP1-E988Q mono-Glu/Asp modification in the absence of HPF1 (left lane).

This view is strengthened by other data in (Bonfiglio et al., Cell, 2020) Supplementary Figure S5, in which they show that both AbD33204 and AbD33205 bind *in vitro* auto-modified PARP10, PARP14 and PARP15 (which are not thought to modify serines), but that AbD33204 often prefers these other modifications over serines, whereas AbD33205 recognizes both. Similarly, Sup. Fig 7 in (Weixler et al, Life Sci Alliance, 2022) shows that AbD33205 (also called HCA355) can recognize PARP10 automodification. Therefore, one cannot draw any conclusions in terms of the ADPr linkage using these reagents alone.

We have improved the way these reagents are referred to in the results section (lines 90-101) and in the figure legends, and included a paragraph in the discussion section about the nature of the IFN γ -induced ADPr, in which the above point is mentioned (lines 247-263).

3. Fig. 1C: the authors state that H₂O₂-mediated nuclear ADP-ribosylation targets Ser which is why the signal doesn't diminish upon hydroxylamine treatment. However, several studies in the literature show that H₂O₂ can indeed generate Glu/Asp-ADPr modifications (e.g., Zhang et al., *Nature Methods*, 2013). The authors should determine if ARH3, a known Ser-ADPr hydrolase, removes this signal. The authors should also use a secondary method to confirm Glu/Asp-ADPr modification in their IF images, for example, showing that Nsp3 macro can remove the signal.

Thank you for this comment, which again raises an interesting point and allows us to reflect on current literature in the field. First, we would like to point out that this experiment was not designed to answer the question of relative proportions of Ser vs Glu/Asp modification by PARP1, which indeed is not the subject matter of this manuscript. Given that the peroxide-induced ADPr signal is known to persist in ARH3 KO cells (e.g. Hanzlikova et al. *Nat Commun*, 2020), and serine modification has been shown by multiple groups to be the predominant form of DNA damage-induced ADPr (e.g. papers by I. Ahel, I. Matic and M. Nielsen groups), there is good evidence that peroxide should induce predominantly hydroxylamine-resistant ADPr. Therefore, the peroxide control sample was merely included to show that hydroxylamine treatment does not prevent detection of ADPr by immunofluorescence in a non-specific manner. Indeed, we have now quantified the effects of hydroxylamine treatment on the immunofluorescence signals - both on the PARP1-induced nuclear signal in response to peroxide treatment, and on the cytoplasmic signal induced by IFN γ (new Figure 1D). These data show that the nuclear PARP1-dependent signal is virtually unchanged after hydroxylamine treatment, suggesting that it is predominantly linked to proteins via hydroxylamine-resistant linkages. Importantly, this quantification confirms that the IFN γ -induced signal is completely removed by hydroxylamine. Corroborating this result by another method, we have incubated cell lysates with hydroxylamine and subsequently detected protein-ADPr by western blotting using the recently-developed AbD43647 reagent (new Fig. EV1A). In agreement with our immunofluorescence data, the IFN γ -induced signal was completely hydroxylamine sensitive.

Regarding removal of the IFN γ -induced signal by the Nsp3 macrodomain, this has already been demonstrated in our previous publication (Russo et al., *J Biol Chem*, 2021), where we showed that the IFN γ -induced immunofluorescence signal is indeed sensitive to expression of WT, but not catalytically inactive, SARS-CoV-2 Nsp3 macrodomain.

We have softened the sentence on PARP1 amino acid specificity (lines 96-97), described the new western blot experiment (line 98-101), emphasized our previous observations regarding macrodomain sensitivity of this signal (lines 86-88) and discussed the hydroxylamine sensitivity of the IFN γ -induced ADPr in a new paragraph in the discussion section (lines 247-263).

4. Fig 3: The authors should confirm the PARP14 nuclear signal using PARP14 siRNAs.

Thank you for this suggestion. This experiment was presented originally as Sup. Fig. 2A (now Fig. EV3A), but may have been missed by the reviewer because of an unfortunate mislabelling of the images. Indeed, this mislabelling was spotted by this reviewer (minor comment 2 below), and has now been corrected. The corrected Figure EV3A shows that the nuclear PARP14 signal is not sensitive to siPARP14 treatment, whereas the IFN γ -induced cytoplasmic signal is. Therefore, the nuclear signal detected by this antibody is predominantly non-specific. This is now more explicitly mentioned in the manuscript text (line 106). However, we do observe some PARP14 in the nucleus using subcellular fractionation followed by western blotting (Fig. 4H), suggesting that a fraction of PARP14 may be nuclear.

5. Fig. 4/Sup. Fig 3: The authors should determine if the PARP14 protein stability comparison between WT and DTX3L is statistically significant.

Thank you for spotting this oversight. We have performed the statistical analysis of the quantification in Sup. Fig. 3C (now Fig. EV4C) and the difference in PARP14 protein stability between WT and DTX3L KO cells is indeed statistically significant.

6. Fig 4f: Why are P14 levels lower with MG132/CQ?

Thank you for this important comment, which somewhat indirectly led us to perform a crucial experiment during the revision process. We were also surprised by this reduction of PARP14 levels. To address the possibility that this was caused by an effect of MG132 on IFN γ signalling, and in response to another reviewer comment (reviewer 3, comment 4), we performed an improved experiment, increasing the MG132 treatment time to 24h and probing for additional factors such as LC3, Ubiquitin and phospho-STAT1. In this new experiment, which now replaces the old Fig. 4F, the reduction of PARP14 levels after MG132 treatment is even more evident, and we also see a similar effect on DTX3L protein levels (quantified in Fig. EV4F). Oddly, MG132 substantially increased STAT1 phosphorylation, which is somewhat inconsistent with the reduced PARP14 and DTX3L protein levels. Therefore, we interpret these effects as secondary consequences of the impaired protein degradation and/or ubiquitin turnover induced by long-term proteasome inhibition by MG132. To reduce this confounding effect, we performed a new experiment (new Fig. 4G), in which MG132 treatment was shortened, and only added after the IFN response was induced. Under these conditions, we identified a mild effect of MG132 on PARP14 protein stability, indicating that there is an involvement of the proteasome on PARP14 degradation in the absence of DTX3L. This would provide a rationale for how DTX3L loss could impact PARP14 protein levels and adds important new information to the manuscript during the revision process. We have discussed this new data in lines 185-199 and 269-279.

7. Does knockdown of DTX3L alter PARP14 cytoplasmic puncta formation or nuclear protein levels? This can be assessed by IF or biochemical fractionation.

Thank you for these questions. The suggested biochemical fractionation was presented in the original submission in Fig. 4G (now Fig. 4H). This experiment shows that the reduction in PARP14 protein levels in PARP9 or DTX3L KO cells affects both the cytoplasmic and nuclear PARP14 pools to a similar extent, indicating that loss of PARP9 or DTX3L does not affect PARP14 translocation to/from the nucleus. Interestingly, this is in contrast to the effects of DTX3L or PARP9 deletion on their reciprocal nuclear localization, with DTX3L KO impairing PARP9 localization to the nucleus and vice-versa.

We have now performed the suggested immunofluorescence experiment (pasted below), but have decided not to include this data in the manuscript. Although the cytoplasmic PARP14 puncta are clearly induced by IFN γ treatment, co-localize with ADPr and are sensitive to siPARP14 treatment (Fig. 3A and EV3A), this antibody also detects a non-specific nuclear signal (Fig. EV3A) and a diffuse cytoplasmic signal (Fig. 3A and below). These signals interfere with accurate quantification of the cytoplasmic PARP14 puncta, and we are therefore not confident enough that the below quantification represents the true effect of DTX3L KO on PARP14 localization.

Despite the caveats discussed above, the above result would suggest that although DTX3L KO cells have overall lower PARP14 levels (Fig.4), the loss of DTX3L does not substantially prevent PARP14 from localizing to cytoplasmic puncta.

8. Given the MG132/CQ studies, is it possible that the loss of DTX3L drives PARP14 into "insoluble" aggregates due to excessive auto-mono-ADP-ribosylation of PARP14

Thank you for raising this intriguing idea. This would certainly be a possibility for western blot experiments in which samples are first lysed in milder buffers such as RIPA buffer, followed by a centrifugation step (in which such aggregates could be lost), prior to protein denaturation in Laemmli Buffer. However, we routinely lyse cells directly in hot Laemmli Buffer, which, due to the high SDS content and temperature, should immediately solubilize and denature all proteins, including potential aggregates. Therefore, the likelihood that protein aggregates were lost during our sample preparation procedure is minimal.

9. Fig. 5: the Af1521 pulldown doesn't show direct mono-ADP-ribosylation. Can the authors show this by IP of DTX3L and probing from mono-ADPr in the presence and absence of a PARP14 inhibitor?

Thank you for this suggestion. The Af1521 pulldown methodology is widely used for isolation of ADP-ribosylated proteins or peptides, and we have included a mutant Af1521 macrodomain construct in our workflow to increase the specificity of the detected interactions. Having said that, we agree that an orthogonal method to demonstrate DTX3L ADP-ribosylation by PARP14 adds weight to this observation. During the revision process, we have made several attempts to IP endogenous DTX3L and detect its ADP-ribosylation by western blotting, as suggested. However, the endogenous IP procedure was not efficient enough to allow reliable detection of IFN γ -induced ADP-ribosylation of DTX3L. However, we have shown, using fluorescent western blotting, that a mono-ADPr-specific reagent (AbD43647) detects a band that overlays with the DTX3L band (Fig. 5B), and that this signal is induced by IFN γ treatment and is lost upon PARP14 inhibition. Although somewhat circumstantial, this is a good indication that DTX3L is ADP-ribosylated by PARP14 in response to IFN γ signalling. Additionally, the accompanying manuscript by the Ahel group shows that recombinant PARP14 can ADP-ribosylate DTX3L *in vitro*. To soften the statements in the relevant section, we have replaced "ADP-ribosylation" with expressions like "Af1521 binding" or "Af1521 interaction" and softened "indicates" to "suggests" (lines 216-236)

Other minor comments:

1. Page 1, line 27: reference for PARP7 is missing.

We believe the reviewer may be referring to page 1, line 37. In this sentence, we use the expression "PARP7 through PARP14" to refer to all members of the ADP-ribosyltransferase family that are IFN-responsive (PARP7,8,9,10,11,12,13 and 14), as recently reviewed in the cited reference (Hoch, Biochem Soc Trans, 2021). Primary research papers indicating that PARP7 is IFN-responsive are cited in this review, and have not been included here, as this would lead to the requirement of additional citations for each of the other members of the family as well. The sentence was amended to clarify this point (line 39).

2. Sup. Fig. 2: labeled incorrectly-should be PARP14.

Thank you for identifying this mistake, which has now been corrected in the revised Fig. EV3A. This error may have unfortunately affected the interpretation of the figure by this reviewer (major comment 4 above).

3. The authors should explain why they switched to RPE-1 cells for certain experiments.

Thank you for this comment. We have previously shown (Russo et al, J Biol Chem, 2021) that IFN γ -induced ADPr can be observed in both A549 and RPE1 cells, so this phenomenon is not limited to particular cell lines, which allows confirmation of results between these cell lines, adding robustness to the study. As A549 is derived from a tumor, while RPE1 is a normal diploid cell line, this also minimizes potential effects of the oncogenic process on our observations. The A549 cells expressing the Nsp3 macrodomain, and the PARP9 and DTX3L KO cells in the RPE1 background, were generated and characterized in our previous publication (Russo et al, J Biol Chem, 2021), so experiments requiring these particular genetic manipulations were performed in these validated cell lines. For the revision, we actually added another cell line (HeLa), as we were unable to transfect the rather large YFP-PARP14 construct into RPE1 or A549 cells. In HeLa, we could also observe an induction of IFN γ -induced ADP-ribosylation in cytoplasmic inclusions, and the localization of PARP9, DTX3L and PARP14 to this structure, confirming the robustness of our results in an additional cellular background. Sentences were added in the results section (lines 86-88 and 128-130) to clarify these points.

Referee #2

Riberio et al. show that addition of IFN-gamma results in increased cellular ADP-ribosylation (ADPr) as detected a microscopy-based assay. They show that this increased ADPr signal is dependent on the catalytic activity of PARP14 and can be reduced by overexpression of a macrodomain (Mac1) from SARS-CoV-2. They further show that PARP14 activity is regulated by the PARP9/DTX3L complex, although many of the mechanisms for this remain to be determined.

There are several interesting aspects to this paper. The authors provide clear data on the requirement for PARP14 in the IFN-induced increase in ADPr signal in their microscopy assay. In addition, they clearly show that PARP14 activity is regulated by PARP9/DTX3L. Both of these pieces of data further support the model that has been emerging from their work and others that PARP14, PARP9, and DTX3L are important regulators of the IFN-gamma response. However, in several places, the authors overinterpret their data, drawing conclusions that are not supported by the data, which weakens enthusiasm for the paper. Detailed comments are below:

Major concerns:

1. Many of the results are dependent on a single microscopy assay. Based on the western blot data shown in Figure 5B (especially the slice shown in the right panel), there is little to no increase in ADPr signal upon IFN-gamma induction, which calls into question the microscopy data. This same inconsistency across assays and/or antibodies

is present in the co-submitted manuscript from Kar et al. Can the authors compare antibodies in both western blot and fluorescence assays to be fully transparent about when there is an IFN-gamma-induced increase in ADPr? Detection of this signal in an antibody-specific way does not discount its existence, but it would be useful for future studies for the authors to be clear about when this can be observed and when it can not and even provide some speculation about the reasons for those differences.

Thank you for raising this important point. We have been able to detect IFN γ -induced ADPr by immunofluorescence in a cytosolic punctate pattern using all four different anti-ADPr detection reagents we tested so far, all of which are shown in this manuscript: pan-ADPr (Millipore- originally from the Kraus lab), pan-ADPr eAf1521-Fc (Hottiger lab), mono-ADPr AbD33204 and mono-ADPr AbD33205 (both BioRad, originally from the Matic lab). This observation has been independently confirmed by several groups, including in the accompanying manuscript by the Ahel group.

The only reagent that we have been able to quite reliably detect IFN γ -induced ADPr is the recently developed AbD43647 reagent (BioRad, Matic group – Longarini et al, Mol Cell, 2023). However, as shown more clearly now in new western blotting data included in the revision (Fig. EV1A, EV2C), while this reagent detects several bands that are clearly induced by IFN γ treatment, there are also a number of quite prominent bands that are insensitive to IFN γ treatment. Similar observations were made by the Ahel group in the accompanying manuscript. We can speculate on two reasons why the IFN γ -induced ADPr signal is easier to detect by immunofluorescence staining than by western blot. One is that IFN γ -induced ADPr may also target RNA (Suskiewicz et al, NAR, 2023), which would likely be detectable by IF, but not WB. The other potential reason is that Glu/Asp-targeted ADPr is known to be a labile modification (Weixler et al, Life Sci Alliance, 2022; Tashiro et al, J Am Chem Soc, 2023), and may be partially lost during sample preparation for western blots.

Additionally, IFN γ -induced ADPr is likely much weaker than the DNA damage-induced signal catalysed by PARP1, which most researchers in the field are most familiar with. Therefore, it is perhaps unsurprising that the IFN γ -induced signal is much more sensitive to differences in sample preparation or on poorly understood differences between ADPr detection reagents. We agree that there is a need for a better understanding of the specificities of these reagents and their “real-world” applicability for different methodologies, as recently highlighted by (Weixler et al, Life Sci Alliance, 2022). We are preparing a manuscript in which we describe our experience with these reagents in detail, which we hope will be of use to other researchers in the field.

A paragraph discussing some of these points was added in the discussion section (lines 247-263)

2. Again for full transparency, it would be appropriate for the authors to show (potentially as supplementary materials) full western blots rather than the single slices as are shown in Figure 3D or 5B. Analysis of the entire blot, rather than the slice that the authors want to focus on, will allow the reader to interpret the data.

Thank you for this comment. As part of the revision process, we have submitted raw data for all figures of the manuscript, as per EMBO J policy. For many of these, the full molecular weight range is not available, as we routinely crop membranes to the expected molecular weight range of the target protein, to allow simultaneous detection of multiple proteins on the same membrane. However, we have included new data during the

revision (Fig. EV1A and EV2C), which show a full molecular weight range mono-ADPr western blot.

3. The authors do not formally show that the ADPr signal that they detect is directly catalyzed by PARP14. Given all of the cross-regulation of PARPs, and the dependence of these enzymes on each other (some of which is nicely shown in this paper), it would be more appropriate to be interpret these results as indicating that PARP14 activity is required for this increase in ADPr signal rather than the premature assertion that it is "catalysed by PARP14".

Thank you for raising this point. While the reviewer is correct in pointing out that we cannot categorically say that PARP14 is the only enzyme catalysing all of the IFN γ -induced ADPr, we have several pieces of evidence that argue that PARP14 is the main enzyme responsible for this modification. We show that inhibiting PARP14 catalytic activity completely prevents all of the detectable IFN γ -induced ADPr (Fig 2D), which would be unlikely to occur if PARP14 were a regulator of another enzyme. For example, loss of PARP9 or DTX3L, which we here show to be regulators of PARP14, only partially affects IFN γ -induced ADPr (Russo et al, J Biol Chem, 2021). The new data showing that efficient depletion of PARPs 7 to 13 does not impair IFN γ -induced ADP-ribosylation also argues against the involvement of these IFN-induced ADP-ribosyltransferases in catalysing this modification. Taken together, we believe that the most parsimonious interpretation of our data is that PARP14 itself is the enzyme catalysing the majority of the IFN γ -induced ADPr. Nonetheless, we changed the title of the manuscript to suppress this expression, and reserved this statement for the discussion section, using expressions such as "promoted" instead of "catalysed" and "PARP14-dependent" instead of "PARP14-catalysed" elsewhere.

4. The assertion that PARP14 and DTX3L ADP-ribosylation are the "first cellular targets of the viral macrodomain" is unsupported by their data and the literature. Many manuscripts have reported that overexpression of the SARS-CoV-2 macrodomain (Mac1) is able to reverse ADP-ribosylation on a variety of cellular targets. It is unclear how these results are different from those other overexpression data, thus these claims of novelty/importance throughout the paper should be removed.

Thank you for this comment. We are unaware of manuscripts reporting that overexpression of the SARS-CoV-2 macrodomain in cells leads to the hydrolysis of ADP-ribose modification of a defined cellular target. In search of manuscripts reporting this activity, we conducted a careful review of the literature. As shown in the table below, several groups have demonstrated that recombinant SARS-CoV-2 Mac1 can hydrolyse ADPr modification of human protein targets in biochemical assays, but we failed to find references showing this activity inside human cells. However, there is evidence by the Leung group showing that the CHIKV macrodomain hydrolyses ADPr modification of human G3BP1 in cells (Jayabalan et al, PNAS 2021), and the Verheugd group demonstrated a similar cellular activity of CHIKV macrodomain towards ADP-ribose modification of the viral nsP2 protein (Eckei et al, Sci Rep 2017). In a recent report by the Ahel group, SARS-CoV-2 Mac1 overexpression was shown to reduce ADPr modification of several bands on western blots induced by PARP14 overexpression, without direct identification of the target proteins (Dukic, Sci Adv 2023).

We have revised the manuscript to replace “viral macrodomain” with “coronavirus macrodomain” where appropriate. Also, to reduce emphasis on this point, we have removed the novelty claim from most sentences, reserving it to our final model (Fig. 5D and lines 341-344).

Reference	Viral hydrolase	Host target	Experimental model
Saikatendu et al. Structure 2005 PMID: 16271890	SARS-CoV	None	
Putics et al. J Virol 2005 PMID: 16188975	HCoV-229E	None	
Egloff et al. J Virol 2006 PMID: 16912299	SARS-CoV SFV HEV	None	
Eriksson et al. J Virol 2008 PMID: 18922871	MHV	None	
Malet et al. J Virol 2009 PMID: 19386706	CHIKV VEEV SINV SARS-CoV SFV	None	
Parvez Gene 2015 PMID: 25870943	Hepatitis E	None	
Fehr et al. J Virol 2015 PMID: 25428866	MHV	None	
Li et al. J Virol 2016 PMID: 27440879	HEV SFV VEEV SARS-CoV HCoV-229E	PARP1, PARP5, PARP10, PARP15 catalytic domain	Biochemical reactions with recombinant proteins
Fehr et al. mBio 2016 PMID: 27965448	SARS-CoV	PARP10 catalytic domain	Biochemical reactions with recombinant proteins
McPherson et al. PNAS 2017 PMID: 28143925	CHIKV	PARP10 catalytic domain	Biochemical reaction with recombinant proteins
Eckei et al. Sci Rep 2017 PMID: 28150709	CHIKV VEEV FIPV SINV ONNV HEV	-PARP10, PARP14, PARP15 catalytic domain (auto) -NEMO (PARP10 trans) -PARP10 automodification	Biochemical reaction with recombinant proteins GFP-PARP14 macro2/3 pulldown of HA-PARP10. Co- express CHIKV macro, lose interaction

			IP HA-PARP10, blot ADPr. Co-express CHIKV macro, lose ADPr signal
Abraham et al. PNAS 2018 PMID: 30322911	CHIKV	None	
Grunewald et al. Virology 2018 PMID: 29199039	MHV PEDV SARS-CoV, MERS-CoV	Viral Nucleocapsid protein	Detect ADPr on N protein: IP N protein, blot ADPr IP ADPr, blot N expression of NSP3 macrodomain does NOT hydrolyse modification
Grunewald et al. PLoS Pathog 2019 PMID: 31095648	MHV	None	
Abraham et al. mBio 2020 PMID: 32047134	SINV	PARP10 catalytic domain	Biochemical reaction with recombinant proteins
Rack et al. Open Biol 2020 PMID: 33202171	SARS-CoV-2	PARP14 WWE-Cat (auto) or PARP14 macro2 or macro3 (trans)	Biochemical reaction with recombinant proteins
Alhammad et al. J Virol 2021 PMID: : 33158944	SARS-CoV-2 MERS-CoV SARS-CoV	PARP10 catalytic domain	Biochemical reaction with recombinant proteins
Jayabalan et al. PNAS 2021 PMID: 33547245	CHIKV	G3BP1	overexpress GFP-G3BP1 + Flag-CHIKV macrodomain (or cat dead), do GFP IP, blot for ADP-ribose. Macrodomain expression reduces G3BP1 ADPr
Voth et al. J Virol 2021 PMID: 34011547	MHV	None	
Russo et al. J Biol Chem 2021 PMID: 34358560	SARS-CoV-2	No defined target	Overexpression of SARS-CoV-2 macrodomain reduces ADP-ribosylation in IF
Schuller et al. Sci Adv 2021 PMID: 33853786	SARS-CoV-2	None	
Roy et al. Antiviral Res 2022 PMID: 35598780	SARS-CoV-2	PARP10 catalytic domain	Biochemical reaction with recombinant proteins
Dasovich et al. ACS Chem Biol 2022 PMID: 34904435	SARS-CoV-2 CHIKV	PARP10 catalytic domain GFP-G3BP1 from cells	Biochemical reaction with recombinant proteins IP GFP-G3BP1 from cells, incubate with recombinant macrodomain, blot ADPr

Aguilar et al. J Virol 2022 PMID: 35297669	SARS-CoV SINV	PARP12 catalytic domain	Biochemical reaction with recombinant proteins
Sherrill et al. Bioorg Med Chem 2022 PMID: 35597097	SARS-CoV-2	PARP10 catalytic domain	Biochemical reaction with recombinant proteins
Kerr et al. J Virol 2023 PMID: 37695054	MHV	None	
Alhammad et al. PNAS 2023 PMID: 37607224	SARS-CoV-2	None	
Taha et al. PLOS Pathog 2023 PMID: 37651466	SARS-CoV-2	PARP10 catalytic domain	Biochemical reaction with recombinant proteins
Kumar et al. Pathogens 2023 PMID: 37242388	CHIKV	Aedes Tankyrase	Biochemical reaction with recombinant proteins
Krieg et al. Cell Mol Life Sci 2023 PMID: 36840772	CHIKV	CHIKV nsP2 trans-ADPr by PARP10, PARP12, PARP14 and PARP15	Biochemical reaction with recombinant proteins IP Nsp2-GFP from cells, incubate with recombinant CHIKV macrodomain, blot ADPr. Recombinant macrodomain hydrolyses modification in lysate
Gahbauer et al. PNAS 2023 PMID: 36598939	SARS-CoV-2	None	
Dukic et al. Sci Adv 2023 PMID: 37703374	SARS-CoV-2	- PARP14 WVEcat (auto) - PARP14 MD3 (trans) - ADPr-RNA or DNA - No defined target	Biochemical reaction with recombinant proteins Co-express YFP-PARP14 MD1 mut + SARS-CoV-2 macrodomain, anti-ADPr blot on cell lysate, expression of macrodomain reduces ADPr on many unidentified bands
Chea et al. biorxiv 2023 PMID: 36945431	SARS-CoV-2	None	
Kerr et al. biorxiv 2024 PMID: 38260573	MERS-CoV SARS-CoV-2	PARP10 catalytic domain	Biochemical reactions with recombinant proteins
Stoll et al. J Virol 2024 PMID: 38289106	Rubella virus	PARP10 catalytic domain	Biochemical reaction with recombinant proteins

Zhu et al. NAR 2024 PMID: 38000390	SARS-CoV-2	ADPr-RNA and Ub-ADPr-RNA	Biochemical reaction with recombinant proteins
---	------------	--------------------------	--

5. A recent paper (<https://pubmed.ncbi.nlm.nih.gov/37695054/>) showed that a coronavirus Mac1 mutant virus is inhibited in vivo in a PARP12-dependent manner. This paper should be cited, rather than only citing work that implicates viral macrodomain activity as being important in the context of PARP14.

Thank you for this very pertinent comment. We have amended sentences in the introduction and discussion to refer to the role of PARP12 in counteracting the coronavirus macrodomain and cited the relevant paper (lines 67-69 and 338-340)

6. The antibody used for PARP14 in Figure 3 does not appear to be entirely specific for PARP14 given how little change there is in the overall signal relative to the western blot data (e.g. Figure 3C). The authors need to either their microscopy signal with a siRNA or KO of PARP14. signal remains in the PARP14 KO cells. Do the authors have another way to support the claim that PARP14 co-localizes with ADPr signal in those cells?

Thank you for this comment. As discussed in the above response to reviewer 1, comment 4, the suggested siRNA experiment to determine the specificity of the PARP14 antibody was already shown in the original submission (Sup. Fig. 2A), but was unfortunately mislabelled. In this experiment (now shown in Fig. EV3A), we demonstrate that the nuclear signal detected by our PARP14 antibody is non-specific, while the IFN γ -induced cytosolic puncta are sensitive to siPARP14. The sentence in line 124/125 was amended to convey this more clearly.

To corroborate that PARP14 co-localizes both with ADPr and with PARP9/DTX3L, we have now performed an experiment using YFP-tagged PARP14 (new Fig. 3C and Fig. EV3E). This experiment shows that YFP-PARP14 also extensively co-localizes with PARP9, DTX3L and IFN γ -induced ADPr.

Other concerns:

1. In Figure 1, do the authors know that hydroxylamine treatment does not impact other aspects of IFN-gamma signaling? Can they show that STAT1 phosphorylation or PARP14 upregulation is intact in hydroxylamine-treated cells?

Thank you for this question. In these experiments, the hydroxylamine treatment occurs after cells have been fixed and permeabilised (for IF experiments, Fig. 1C-D) or after the cells have been lysed (for WB experiments, Fig. EV1A). Therefore, hydroxylamine is just a chemical processing of the samples after the cells are dead, without any possibility for a cellular response to this treatment. We have amended the figure legends and methods section to clarify this point (lines 420 and 680).

2. In the text associated with Fig 4A and Supplementary Figure 3A, the authors claim that PARP9 silencing led to a 'striking reduction in PARP14 levels'. Neither of these figures support that claim. Visually, Figure 4A shows no difference. Quantitatively, Supplementary Figure 3A does not show a significant reduction. Demonstratively false statements such as these reduce the confidence in conclusions drawn in the paper.

Thank you for raising this point. This was an unfortunate mistake. The "striking reduction" was meant to refer to the siDTX3L sample, without extending to the siPARP9 sample,

but the sentence as written did indeed convey this erroneous idea. We have amended the sentence (line 158) to more accurately describe these results. However, we show in other data (Fig. 4E and 4H) that PARP9 KO does indeed lead to a substantial reduction in PARP14 levels, similar to DTX3L KO. Therefore, the milder effect of siPARP9 on PARP14 levels shown in Fig. 4A is likely to reflect a reduced efficiency of the PARP9 depletion in this experiment compared to siDTX3L.

3. IFN-gamma is not the only type of IFN. The authors should be more specific when referring to "IFN" when they only test IFN-gamma.

Thank you for this suggestion. We have shown previously (Russo et al, J Biol Chem, 2021) that ADP-ribosylation can be induced by recombinant IFN α , IFN β and IFN γ , as well as by transfection with poly(I:C). As the ADPr signal is always strongest using IFN γ , we used it for most experiments in this study. While it is likely that many of the observations shown here also extend to type I IFN signalling, we have revised the manuscript to replace "IFN" with "IFN γ " where appropriate and mentioned our previous observations more clearly in some places (lines 48 and 83).

Referee #3

This is an interesting manuscript that seeks to understand the role of DTX3L/PARP9 in modulating PARP14 levels/biochemical output. The biological interest in this question stems in part from the observation that PARP14 appears to be responsible for the bulk of ADP-ribosylation that occurs in response to IFN treatment, which seems logical given that PARP14 itself is induced robustly by IFN. Unfortunately, the authors never answer the question of how DTX3L contributes to PARP14 levels/activity in the cell. The argument for a post-transcriptional mechanism is not well-founded using the data provided. The finding that PARP14 binding to AF1521 beads can be reduced by over-expression of the CoV-2 Nsp3 macrodomain would be intriguing if Nsp3 modulates PARP14 ADP-ribosylation of cellular targets relevant to a viral response. As presented, the Nsp data might also be accounted for by competing with AF1521 for binding ADP-ribosylated substrates including auto-modified PARP14.

Specific

comments

Figure 1: The fluorescence imaging is intended to show that IFN signaling induces mono-ADP-ribosylation on acidic residues in target proteins. The probe used to draw this conclusion, eAF1521, is a pan-ADP-ribose detection reagent that binds both mono- and poly-ADP-ribosylated substrates, so it is hard to know what fraction of each is induced by IFN. The authors have access to mono-specific reagents (Fig. 1A, B), but as far as I can tell the mono-specific reagents are not used in the paper beyond the first figure. Presumably IFN-induced ADP-ribosylation on hydroxylamine-sensitive sites is a novel observation; the impact of this observation would be enhanced by quantification of the fluorescent images, and by parallel biochemical analysis (blotting; quantification).

Thank you for these comments. The conclusion that at least part of the IFN γ -induced modification is mono-ADPr is drawn from the data using mono-ADPr specific reagents (AbD33204 and AbD33205) (Fig. 1A and 1B), not from data using the eAF1521 pan-ADPr reagents. These figures show not only that two different mono-ADPr reagents can detect IFN γ -induced ADP-ribosylation, but also that the mono-ADPr signal extensively co-localizes with the pan-ADPr signal. Therefore, these signals are specific and the same structure can be detected by all of these reagents. As the two mono-ADPr reagents only

recently became commercially available, most of the study was performed using pan-ADPr reagents (eAF1521-Fc and the related pan-ADPr reagent from Millipore). However, given that these signals co-localize (as shown in Fig. 1A and 1B), the remaining figures are reporting on the same structure/event. Furthermore, all of the western blotting data (Fig. EV1A, EV2C and Fig. 5B) were performed using a third mono-ADPr-specific reagent (AbD43647), further corroborating, by a different method, that the modification is likely to be mono-ADPr. A paragraph on this point was included in the discussion section (lines 247-263)

Regarding the hydroxylamine sensitivity, we have now quantified the immunofluorescence images (new Fig. 1D), which demonstrates that IFN γ -induced ADPr in the cytosol is highly hydroxylamine sensitive, whereas the peroxide-induced nuclear ADPr is not. We have also performed a similar experiment by western blotting, which again shows that the IFN γ -induced ADPr is sensitive to hydroxylamine incubation (new Fig EV1A). A sentence on this new result was included in lines 98-100.

Figure 2: The authors used a focused siRNA screen and found that PARP14 mediates IFN-induced ADP-ribosylation in cells - which also is sensitive to the PARP14i from Ribon Inc. The data would be strengthened with a parallel biochemical analysis since the spots detected in the microscope are of unknown etiology, spare the fact they are induced by IFN.

Thank you for these suggestions. We have only recently succeeded in reliably detecting IFN γ -induced ADPr using western blots, as discussed in more detail in the response to reviewer 2, comment 1 above. We have performed the suggested experiment using this western blotting method, which nicely confirmed the siRNA screen (new Fig EV2C, described in lines 111-113). The effect of PARP14 inhibitor on this western blot signal is shown in Figure 5B

Figure 3: The authors look for interactions between PARP14 and DTX3L/PARP9 after inferring both localize to small cytoplasmic foci labeled by eAF1521. Two points here - the authors should consider imaging epitope-tagged PARP14 and PARP9 in the same cells if the basic question is one of co-localization, since this will enable asking the question without the complication of changes in PARP14 expression induced by IFN. Second, the PARP14 immunoprecipitation is so inefficient I would be very cautious about interpreting this negative result. Overall, not much is learned from this figure since it is based on a negative result.

Thank you for these suggestions, which strengthened the manuscript. We have now performed an experiment expressing YFP-PARP14 in cells, which shows that YFP-PARP14 extensively co-localizes with both endogenous PARP9 or DTX3L, as well as IFN γ -induced ADPr (new Fig. 3C and new Fig. EV3E). These experiments had to be performed in HeLa cells due to the low transfection efficiency of the YFP-PARP14 construct in RPE1 or A549 cells. However, this allowed us to extend our findings to an additional cell line, since IFN γ treatment of HeLa cells also induced ADP-ribosylation in a cytoplasmic structure containing PARP9, DTX3L and PARP14, as already seen in RPE1 and A549. These results are presented in lines 129-133.

We agree that the endogenous IP (original Fig. 3C), was not efficient enough to allow a definitive conclusion and have replaced this figure with a GFP-trap experiment using the same YFP-PARP14 construct (new Fig. 3E). Using this more efficient IP methodology and overexpressed protein, we were able to detect an interaction between YFP-PARP14 and endogenous DTX3L, described in lines 143-148. Indeed, the domains likely to

mediate this interaction have been identified in (Saleh et al, J Mol Biol, 2024), which was published during the revision of this manuscript. This result is now presented in lines 143-148 and discussed in lines 297-302.

Figure 4: The key panels in this figure (4C, D) attempt to address how DTX3L affects PARP14 expression. The data as presented do not permit a clear assessment. The authors suggest a transcriptional effect of DTX3L was ruled out, but the "non-significant" difference in RPE-1 WT vs. RPE-1 DTX3L KO might be explained by replicate variance. The CHX experiment addressing PARP14 half-life is not readily interpretable without quantification. Ideally the PARP14 half-life measurements will be made after IFN treatment given that PARP14 activity seems most relevant in the context of IFN signaling. The MG132 and CLQ data as presented (neg results) without positive controls cannot be used to rule out proteasome and autophagy involvement; it would be appropriate to test if/how the protein half-life is affected by these treatments since the result seems surprising. The biochemical fractionation (4G) is well done but I can't judge if this contributes to the understanding.

Thank you for raising these important points, which led to a series of experiments that helped shed light on this aspect of the manuscript. Regarding the transcriptional effect of DTX3L KO on PARP14 mRNA levels, we have added an additional replicate to this experiment (new Fig 4C), in which again the deletion of DTX3L KO had no major effect on PARP14 mRNA levels (see figure below highlighting the data of the 5th replicate in red). Given that we observed a slightly reduced PARP14 mRNA level in DTX3L KO cells under basal conditions, we reanalysed the data by normalizing the IFN γ or poly(I:C)-treated samples to their respective untreated control. This figure (also pasted below) shows that the induction of PARP14 mRNA by interferon signalling is unaffected by DTX3L KO. We chose to keep the current graph, normalized to the untreated WT samples, in order to be fully transparent about the observed biological effects. Further strengthening our observations, this lack of an effect of DTX3L or PARP9 loss on PARP14 mRNA levels was independently confirmed by the Ahel group in the accompanying manuscript and by the Grundy group in the (Saleh et al, J Mol Biol, 2024) paper, which was published during the revision of this manuscript. We changed the wording in line 268 to "this regulation is unlikely to be mediated through transcriptional control" to soften this statement, and added reference to the Saleh paper confirming this observation.

Regarding the cycloheximide experiment (Fig. 4D), the quantification of this data was shown in the originally submitted manuscript in Sup. Fig. 3C (now Fig. EV4C). We have now also added statistical analysis to this data, showing that PARP14 protein half-life is shorter in DTX3L KO cells. Again, a strikingly similar result was observed independently by the Grundy group in the (Saleh et al, J Mol Biol, 2024) paper in different cell lines. We have performed an improved experiment that now replaces Fig. 4F. For this, we have increased the MG132 and chloroquine treatment times to 24h, and probed for additional factors such as LC3, ubiquitin and phospho-STAT1, as suggested. These controls demonstrated that MG132 and CQ were indeed working, but neither of them were able to substantially restore PARP14 protein levels in DTX3L KO cells. Indeed, as discussed in more detail in response to reviewer 1 comment 6, the MG132 treatment surprisingly induced a reduction in PARP14 and DTX3L levels even in WT cells, accompanied by a surprising increase in STAT1 phosphorylation. Therefore, this experiment does not allow any conclusions on the effect of proteasome inhibition on PARP14 protein stability in the DTX3L KO background.

To circumvent this non-specific effect of MG132 on the interferon response, and to determine the PARP14 half-life in interferon-treated cells, as suggested, we performed another experiment, in which the interferon response was first induced for 24h, followed by CHX treatment for up to 8h, in the presence or absence of MG132 (new Fig. 4G and new Fig. EV4G). Under these conditions, we observed again that DTX3L KO reduced PARP14 protein half-life, and we were now able to observe a mild effect of MG132, which delayed PARP14 degradation in the DTX3L KO cells. Therefore, although this effect was somewhat mild, these data suggest that PARP14 could be a target of the proteasome. These data are presented in lines 185-199 and discussed in lines 269-273, including a reference to the Saleh paper.

The relevance of the biochemical fractionation experiment is discussed in response to reviewer 1, comment 7 (above).

Figure 5: The authors show that Nsp overexpression reduces the amount of PARP14 and DTX3L captured on AF1521 beads. Since the question is whether the viral macrodomain hydrolyzes ADP-ribose from these proteins, it would seem prudent for the authors to IP PARP14 and DTX3L separately and probe for ADP-ribose on blots as a more direct test. Otherwise, there are multiple possibilities for how Nsp expression might affect protein recovery on AF1521 beads using cell lysates.

Thank you for this pertinent comment. The Af1521 pulldown methodology is widely used for isolation of ADP-ribosylated proteins or peptides, and we have included a mutant Af1521 macrodomain construct in our workflow to increase the specificity of the detected interactions. Having said that, we agree that an orthogonal method to demonstrate DTX3L ADP-ribosylation by PARP14 adds weight to this observation. During the revision process, we have made several attempts to IP endogenous DTX3L and PARP14, and detect their ADP-ribosylation by western blotting, as suggested. However, the endogenous IP procedure was not efficient enough to allow reliable detection of IFN γ -induced ADP-ribosylation of either DTX3L or PARP14. However, the overexpression of the SARS-CoV-2 macrodomain was shown to largely remove either IFN γ -induced and/or PARP14-dependent ADP-ribosylation, both in our previous publication (Russo et al, J Biol Chem, 2021) by immunofluorescence, and in a recent publication by the Ahel group (Dukic et al, Sci Adv 2023), using western blotting. To soften statements, we have replaced “ADP-ribosylation” with expressions like “*Af1521* binding” or “*Af1521* interaction” and softened “indicates” to “suggests” (lines 216-236). We have also changed the title, abstract and results sections to suppress mention of the macrodomain-dependent hydrolysis of PARP14 and DTX3L ADPr, reserving this statement to lines 341-344 of the discussion, where we present our model. We have rewritten a substantial section of the discussion (lines 312-340) to more clearly discuss previous observations that are consistent with this model, while also mentioning the possibility that this result could be an artefact of competition between Af1521 and Nsp3 macrodomain binding to these proteins, as suggested. (lines 353-360)

Dr. Nicolas Carlos Hoch
University of Sao Paulo
Biochemistry
Av Prof. Lineu Prestes 748
Sao Paulo, Sao Paulo 05508-000
Brazil

30th Apr 2024

Re: EMBOJ-2023-115815R
PARP14 is regulated by the PARP9/DTX3L complex and promotes interferon γ -induced ADP-ribosylation

Dear Nicolas,

Thank you for submitting your revised manuscript to The EMBO Journal. It has now been seen once more by two of the original referees, and I am happy to say that both were generally satisfied with your revisions and responses to the initial comments. However, referee 2 is still concerned about certain interpretations that appear to over-reach the presented data, and suggests that they be modified prior to publication. I am therefore returning the manuscript to you for a final round of minor revision, inviting you to incorporate these suggestions and to provide another brief response letter.

In addition, please carefully address also the following editorial issues in the final version:

- Please reduce the number of keywords on the abstract page to five (ideally choosing broad general terms).
- Please double-check all citations in the reference list, as many of them (e.g., Javed, Leung, Palazzo, Schuller, Chatrin, Delgado...) appear to be still incomplete (lacking page/locator numbers).
- Please rename the Conflict of Interest section into "Disclosure and Competing Interests Statement", in accordance with our updated Guide to Authors (<https://www.embopress.org/competing-interests>)
- As we are switching from a free-text author contribution statement towards a more formal statement based on Contributor Role Taxonomy (CRediT) terms, please remove the present Author Contribution section and instead specify each author's contribution(s) directly in the Author Information page of our submission system during upload of the final manuscript. See <https://casrai.org/credit/> for more information.
- Finally, our routine pre-acceptance image checks revealed an overlap between microscope fields of two panels in different figures, Figure 2A and Figure EV2B - siNT/IFN γ condition. I realize that these are the identical conditions and the images probably taken from the same experiment, but this is not stated in the figure legends. Please clarify, and ideally replace one of the two panels with a non-overlapping field.

I am therefore returning the manuscript to you for a final round of minor revision, to allow you to make the requested presentational and editorial modifications, and upload the revised files. Once we will have received them, we should hopefully be ready to swiftly proceed with formal acceptance and production of the manuscript. I would appreciate if you could send the final version back within one week from now.

With kind regards,

Hartmut

1) Every manuscript requires a Data Availability section (even if only stating that no deposited datasets are included). Primary datasets or computer code produced in the current study have to be deposited in appropriate public repositories prior to

resubmission, and reviewer access details provided in case that public access is not yet allowed. Further information: embopress.org/page/journal/14602075/authorguide#dataavailability

9) Digital image enhancement is acceptable practice, as long as it accurately represents the original data and conforms to community standards. If a figure has been subjected to significant electronic manipulation, this must be clearly noted in the figure legend and/or the 'Materials and Methods' section. The editors reserve the right to request original versions of figures and the original images that were used to assemble the figure. Finally, we generally encourage uploading of numerical as well as gel/blot image source data; for details see: embopress.org/page/journal/14602075/authorguide#sourcedata

At EMBO Press, we ask authors to provide source data for the main manuscript figures. Our source data coordinator will contact you to discuss which figure panels we would need source data for and will also provide you with helpful tips on how to upload and organize the files.

In the interest of ensuring the conceptual advance provided by the work, we recommend submitting a revision within 3 months (29th Jul 2024). Please discuss the revision progress ahead of this time with the editor if you require more time to complete the revisions. Use the link below to submit your revision:

Link Not Available

Referee #1:

The authors have thoroughly addressed all of my comments and suggestions. I recommend publication.

Referee #2:

This manuscript is substantially improved and, along with the accompanying manuscript, will be an interesting addition to the field.

The one place the authors still overinterpret their data and reduce the rigor of their conclusions is with assertion that their data "indicates that the coronavirus macrodomain evolved to disrupt this interaction" (the interaction between PARP9/DTX3L/PARP14) and that PARP14 automodification and modification of DTX3L are "the first cellular targets of the coronavirus macrodomain". Despite their response to my previous comment, there is no evidence in this paper to support these claims. As the authors point out, Dukic et al., *Sci Adv*, 2023, show coexpression of PARP14 with various macrodomains in cells. In these assays, the coronavirus macrodomain reduces the ADPr signal of nearly every band on the western, including the PARP14 automodification band. Importantly, the same widespread reduction of ADPr signal that Dukic et al observe with coronavirus macrodomain is observed upon overexpression of a cellular macrodomain (MACROD1). These data argue that macrodomain activity is promiscuous under these types of overexpression conditions, since presumably MACROD1 has not "evolving to disrupt" PARP14 modification like Riberio et al are arguing the coronavirus macrodomain has. Likewise, Delgado-Rodriguez et al, *Pathogens*, 2023, show that when coexpressed in cells, the coronavirus macrodomain reduces PARP10 automodification to the same degree as the archaeal macrodomain Af1521, which again, has presumably not evolved to disrupt the same processes as the coronavirus macrodomain. Combined, these papers identify both PARP14 and PARP10 as previous targets of the coronavirus macrodomain, and also show these are targets of other overexpressed macrodomains. But more importantly, these papers and papers on other viral macrodomains show that macrodomains that are expressed in isolation are quite promiscuous in their targets, making these assays unsuitable to address the relevant cellular targets of viral macrodomains. These unsupported assertions about coronavirus macrodomain function in Riberio et al's manuscript are not required for the impact of the paper, and serve to only add unsubstantiated claims to the field of viral macrodomain function. It would be much more appropriate to propose this as a possible mechanism of viral macrodomain function, in addition to the possibility that the coronavirus macrodomain removes ADPr from any of the many other proteins that are ADP-ribosylated in the cell.

If the authors feel the need to include statements about the specificity of coronavirus macrodomain targets and function, several additional experiments are needed. For instance, can the authors show that MACROD1 or Af1521 do not have the same effect on PARP14 or DTX3L? Or that the coronavirus macrodomain becomes dispensable for viral replication if the PARP14 and DTX3L modifications are eliminated? The latter is unlikely to be true, given that Kerr et al, *J Virol*, 2023, show that a coronavirus macrodomain mutant is rescued in several (although not all) PARP12 knockout cells and tissues.

UNIVERSIDADE DE SÃO PAULO
INSTITUTO DE QUÍMICA

São Paulo, 3rd May 2024

Dear Dr. Hartmut Vodermaier, *The EMBO Journal*

Enclosed please find a revised version of our manuscript entitled "**PARP14 is regulated by the PARP9/DTX3L complex and promotes interferon γ -induced ADP-ribosylation**", for consideration for publication as an original research article in *EMBO Journal*.

We would like to thank you and the reviewers again for a thorough and careful revision process, which has led to important improvements to the manuscript. Changes made to the manuscript during this minor revision have been highlighted in yellow in the attached file. We have addressed the editorial issues relating to keywords, referencing, conflict statements and author contributions, as suggested. We have uploaded a revised Figure EV2B in which the siNT panels are omitted altogether, given that a representative image of the result for siNT samples in this experiment is already shown in Fig. 2A. The figure legend to EV2B was amended accordingly. Regarding the comment by reviewer 2, we agree with their reasoning, which was now much more clearly expressed. We have made changes to the text in lines 20-22, 335-340 and 345-347 to reflect these ideas.

Sincerely,

Dr. Nicolas Carlos Hoch

Dr. Nicolas Carlos Hoch
University of Sao Paulo
Biochemistry
Av Prof. Lineu Prestes 748
Sao Paulo, Sao Paulo 05508-000
Brazil

8th May 2024

Re: EMBOJ-2023-115815R1
PARP14 is regulated by the PARP9/DTX3L complex and promotes interferon γ -induced ADP-ribosylation

Dear Dr. Hoch,

Thank you for submitting your final revised manuscript for our consideration. I am pleased to inform you that we have now accepted it for publication in The EMBO Journal.

Yours sincerely,

Hartmut Vodermaier
